# Diffusion Models Meet Contextual Bandits

**Imad Aouali**
Criteo AI Lab
CREST, ENSAE, IP Paris
`i.aouali@criteo.com`

## Abstract

Efficient online decision-making in contextual bandits is challenging, as methods without informative priors often suffer from computational or statistical inefficiencies. In this work, we leverage pre-trained diffusion models as expressive priors to capture complex action dependencies and develop a practical algorithm that efficiently approximates posteriors under such priors, enabling both fast updates and sampling. Empirical results demonstrate the effectiveness and versatility of our approach across diverse contextual bandit settings.

## 1 Introduction

A contextual bandit models online decision-making under uncertainty [49]. At each round, an agent observes a context, selects an action, and receives a reward, aiming to maximize cumulative reward by balancing exploitation of high-reward actions and exploration of uncertain ones. However, in large-scale settings (e.g, the number of actions $K$ is large), standard exploration strategies (e.g., LinUCB [11] or LinTS [60]) often become computationally expensive or statistically inefficient. Fortunately, actions in many real-world problems exhibit correlations, enabling more efficient exploration since observing one action can inform the agent about others. Thompson sampling is particularly well-suited for this, as it naturally incorporates informative priors [33] that capture complex action dependencies. Inspired by the success of diffusion models [62, 30], which excel at approximating complex high-dimensional distributions [23, 58], this work leverages pre-trained diffusion models as priors in contextual Thompson sampling.

Precisely, we introduce a framework for contextual bandits with a *diffusion-derived prior*, and develop diffusion Thompson sampling (`dTS`) that is both computationally and statistically efficient. `dTS` achieves fast posterior updates and sampling through an efficient approximation that becomes exact when the diffusion prior and the likelihood are linear. A key contribution, beyond applying pre-trained diffusion models in contextual bandits, is the efficient *computation* and *sampling* of the posterior distribution of a $d$-dimensional parameter $\theta \mid \mathcal{D}$, with $\mathcal{D}$ representing the data, when using a pre-trained diffusion model prior on $\theta$. This is relevant not only to bandits and RL but also to a broader range of applications [19]. Our approximations are motivated by exact closed-form solutions obtained in cases where both the pre-trained diffusion model and the likelihood are linear. These solutions form the basis for our approximations for the non-linear case, demonstrating both strong empirical performance and computational efficiency. Our approach avoids the computational burden of heavy approximate sampling algorithms required for each latent parameter.

Diffusion models have been applied to offline decision-making [6, 36, 65], but their use in online learning has only recently been explored by Hsieh et al. [34] who studied the multi-armed bandit setting, and Kveton et al. [44] who explored a similar direction, with their preprint appearing shortly after the first version of this work. An earlier, deliberately simplified version of our approach, restricted to a linear diffusion model prior, was introduced in Aouali [7]. Here, we extend that formulation to the more realistic and expressive non-linear case. A detailed discussion of related work is provided in Appendix A.

39th Conference on Neural Information Processing Systems (NeurIPS 2025).

## 2   Setting

The agent interacts with a *contextual bandit* over $n$ rounds. In round $t \in [n]$, the agent observes a *context* $X_t \in \mathcal{X}$, where $\mathcal{X} \subseteq \mathbb{R}^d$ is a *context space*, takes an *action* $A_t \in [K]$, and then receives a stochastic reward $Y_t \in \mathbb{R}$ that depends on both the context $X_t$ and the taken action $A_t$.

We focus on the *per-action* (disjoint) setting, where each action $a \in [K]$ is represented by an unknown parameter vector $\theta_a \in \mathbb{R}^d$, so that the reward received in round $t$ is $Y_t \sim P(\cdot \mid X_t; \theta_{A_t})$, where $P(\cdot \mid x; \theta_a)$ is the reward distribution of action $a$ in context $x$. The reward distribution is parametrized as a generalized linear model (GLM) [54]. That is, $P(\cdot \mid x; \theta_a)$ is an exponential-family distribution with mean $g(x^\top \theta_a)$, where $g$ is the mean function. For example, we recover linear bandits when $P(\cdot \mid x; \theta_a) = \mathcal{N}(\cdot; x^\top \theta_a, \sigma^2)$ where $\sigma > 0$, and logistic bandits [24] with $g(u) = (1 + \exp(-u))^{-1}$ and $P(\cdot \mid x; \theta_a) = \mathrm{Ber}(g(x^\top \theta_a))$, where $\mathrm{Ber}(p)$ denotes the Bernoulli distribution with mean $p$. All derivations and algorithms extend naturally to the *shared-parameter* case described in Remark 2.1.

We consider the *Bayesian* bandit setting [59, 33, 55, 28], where the true action parameters $\theta_a$ are assumed to be drawn from a *known* prior distribution. As both the true parameters and the model parameters are sampled from this same prior, we use them interchangeably as a slight abuse of notation. We proceed to define this prior distribution using a diffusion model. The correlations between the action parameters $\theta_a$ are captured through a diffusion model, where they share a set of $L$ consecutive *unknown latent parameters* $\psi_\ell \in \mathbb{R}^d$ for $\ell \in [L]$. Precisely, the action parameter $\theta_a$ depends on the $L$-th latent parameter $\psi_L$ as

$$\theta_a \mid \psi_1 \sim \mathcal{N}(f_1(\psi_1), \Sigma_1),$$

where the *link function* $f_1$ and covariance $\Sigma_1$ are *known*. In particular, the action parameters $\theta_a$ are conditionally independent given $\psi_1$. Also, the $\ell - 1$-th latent parameter $\psi_{\ell-1}$ depends on the $\ell$-th latent parameter $\psi_\ell$ as

$$\psi_{\ell-1} \mid \psi_\ell \sim \mathcal{N}(f_\ell(\psi_\ell), \Sigma_\ell),$$

where $f_\ell$ and $\Sigma_\ell$ are *known*. Finally, the $L$-th latent parameter $\psi_L$ is sampled as $\psi_L \sim \mathcal{N}(0, \Sigma_{L+1})$, where $\Sigma_{L+1}$ is *known*. We summarize this model in Eq. (1) below

$$
\begin{aligned}
\psi_L &\sim \mathcal{N}(0, \Sigma_{L+1}), & &(1)\\
\psi_{\ell-1} \mid \psi_\ell &\sim \mathcal{N}(f_\ell(\psi_\ell), \Sigma_\ell), & &\forall \ell \in [L]/\{1\}, \\
\theta_a \mid \psi_1 &\sim \mathcal{N}(f_1(\psi_1), \Sigma_1), & &\forall a \in [K], \\
Y_t \mid X_t, \theta_{A_t} &\sim P(\cdot \mid X_t; \theta_{A_t}), & &\forall t \in [n].
\end{aligned}
$$

Eq. (1) represents a Bayesian bandit, where the agent interacts with a bandit instance defined by $\theta_a$ over $n$ rounds (4th line in Eq. (1)). These action parameters $\theta_a$ are drawn from the generative process in the first three lines of Eq. (1). In practice, Eq. (1) can be built by pre-training a diffusion model on offline estimates of the action parameters $\theta_a$.

The goal of the agent is to minimize its *Bayes regret* [59], which measures the expected performance across multiple bandit instances $\theta$ that are sampled from the prior,

$$\mathcal{BR}(n) = \mathbb{E}\Big[ \sum_{t=1}^n r(X_t, A_{t,*}; \theta) - r(X_t, A_t; \theta) \Big],$$

where the expectation is taken over all random variables in Eq. (1). Here $r(x, a; \theta) = \mathbb{E}_{Y \sim P(\cdot \mid x; \theta_a)}[Y]$ is the expected reward of action $a$ in context $x$, and $A_{t,*} = \arg\max_{a \in [K]} r(X_t, a; \theta)$ is the optimal action in round $t$. The Bayes regret captures the benefits of using informative priors [33, 32, 8], and hence it is suitable for our problem.

*Remark* 2.1 (**Single shared action parameter**). Our algorithm and analysis also apply to the case where all actions share a single unknown parameter $\theta \in \mathbb{R}^d$. Let $\varphi : \mathcal{X} \times [K] \to \mathbb{R}^d$ be a known feature map, and assume the reward distribution mean is $g\big(\varphi(x, a)^\top \theta\big)$. Then, the diffusion prior in Eq. (1) specializes by replacing the per-action parameters $(\theta_a)_{a \in [K]}$ with a single shared parameter $\theta$:

$$
\begin{aligned}
\psi_L &\sim \mathcal{N}(0, \Sigma_{L+1}), & &(2)\\
\psi_{\ell-1} \mid \psi_\ell &\sim \mathcal{N}(f_\ell(\psi_\ell), \Sigma_\ell), & &\forall \ell \in [L] \setminus \{1\}, \\
\theta \mid \psi_1 &\sim \mathcal{N}(f_1(\psi_1), \Sigma_1), & & \\
Y_t \mid X_t, A_t, \theta &\sim P\big(\cdot \mid \varphi(X_t, A_t)^\top \theta\big), & &\forall t \in [n].
\end{aligned}
$$

This formulation is useful when a shared feature map $\varphi$ is available. In that case, the diffusion model can be pre-trained on parameters $\{\theta_s\}_{s=1}^S$ from previous tasks, and dTS can then be applied to a new task $S+1$ using the pre-trained prior. To avoid clutter, our main exposition focuses on the model in Eq. (1), but all theoretical results and algorithmic components extend naturally to this shared-parameter case, which we also include in some experiments (explicitly noted when applicable).

## 3 Diffusion contextual Thompson sampling

### 3.1 Algorithm

We design a Thompson sampling algorithm that samples the latent and action parameters hierarchically [51]. Let $H_t = (X_i, A_i, Y_i)_{i \in [t-1]}$ denote the history of all interactions up to round $t$, and let $H_{t,a} = (X_i, A_i, Y_i)_{\{i \in [t-1]; A_i = a\}}$ be the history of interactions *with action* $a$ up to round $t$. To motivate our algorithm, we decompose the posterior density $p(\theta_a \mid H_t)$ recursively as

$$p(\theta_a \mid H_t) = \int_{\psi_{1:L}} p(\psi_L \mid H_t) \prod_{\ell=2}^{L} p(\psi_{\ell-1} \mid \psi_\ell, H_t) p(\theta_a \mid \psi_1, H_{t,a}) \, \mathrm{d}\psi_{1:L}. \tag{3}$$

**Hierarchical sampling.** This decomposition induces the following sampling procedure in round $t$. First, draw a sample $\psi_{t,L}$ according to the posterior density $p(\psi_L \mid H_t)$. Then, for each $\ell \in [L] \setminus \{1\}$, draw $\psi_{t,\ell-1}$ from the conditional posterior $p(\psi_{\ell-1} \mid \psi_{t,\ell}, H_t)$. Finally, given $\psi_{t,1}$, draw each action parameter independently from $p(\theta_a \mid \psi_{t,1}, H_{t,a})$ (the $\theta_a$ are conditionally independent given $\psi_1$). This defines Algorithm 1, **d**iffusion **T**hompson **S**ampling (dTS).

**Posterior components via recursion.** To implement dTS, we provide an efficient recursive scheme to express the required posteriors using known quantities. These expressions may not always admit closed forms and can require approximation. The conditional action-posterior can be written as

$$p(\theta_a \mid \psi_1, H_{t,a}) \propto \prod_{i \in S_{t,a}} P(Y_i \mid X_i; \theta_a) \mathcal{N}(\theta_a; f_1(\psi_1), \Sigma_1), \tag{4}$$

where $S_{t,a} = \{\ell \in [t-1] : A_\ell = a\}$ is the set of rounds in which action $a$ was selected. Moreover, let $p(H_t \mid \psi_\ell)$ denote the likelihood of the observations up to round $t$ given $\psi_\ell$. For any $\ell \in [L] \setminus \{1\}$, the conditional latent-posterior is

$$p(\psi_{\ell-1} \mid \psi_\ell, H_t) \propto p(H_t \mid \psi_{\ell-1}) \mathcal{N}(\psi_{\ell-1}; f_\ell(\psi_\ell), \Sigma_\ell),$$

and the top-layer posterior is

$$p(\psi_L \mid H_t) \propto p(H_t \mid \psi_L) \mathcal{N}(\psi_L; 0, \Sigma_{L+1}).$$

All terms above are known except the likelihoods $p(H_t \mid \psi_\ell)$, which are computed recursively. The recursion starts with

$$p(H_t \mid \psi_1) = \prod_{a=1}^{K} \int_{\theta_a} \left[ \prod_{i \in S_{t,a}} P(Y_i \mid X_i; \theta_a) \right] \mathcal{N}(\theta_a; f_1(\psi_1), \Sigma_1) \, \mathrm{d}\theta_a, \tag{5}$$

and for $\ell \in [L] \setminus \{1\}$, proceeds as

$$p(H_t \mid \psi_\ell) = \int_{\psi_{\ell-1}} p(H_t \mid \psi_{\ell-1}) \mathcal{N}(\psi_{\ell-1}; f_\ell(\psi_\ell), \Sigma_\ell) \, \mathrm{d}\psi_{\ell-1}. \tag{6}$$

All posterior expressions above use known quantities $(f_\ell, \Sigma_\ell, P(y \mid x; \theta))$. However, these expressions typically need to be approximated, except when the link functions $f_\ell$ are linear and the reward distribution $P(\cdot \mid x; \theta)$ is linear-Gaussian, where closed-form solutions can be obtained with careful derivations. These approximations are not trivial, and prior studies often rely on computationally intensive approximate sampling algorithms. In the following sections, we explain how we derive our efficient approximations which are motivated by the closed-form solutions of linear instances.

**Algorithm 1** `dTS`: **d**iffusion **T**hompson **S**ampling

---

**Input:** Prior components $\{f_\ell, \Sigma_\ell\}_{\ell=1}^{L+1}$ and reward model $P$.
**for** $t = 1, \ldots, n$ **do**
    Draw $\psi_{t,L}$ according to the posterior density $p(\psi_L \mid H_t)$
    **for** $\ell = L, \ldots, 2$ **do**
        Draw $\psi_{t,\ell-1}$ according to $p(\psi_{\ell-1} \mid \psi_{t,\ell}, H_t)$
    **for** $a = 1, \ldots, K$ **do**
        Draw $\theta_{t,a}$ according to $p(\theta_a \mid \psi_{t,1}, H_{t,a})$
    Select action $A_t = \arg\max_{a \in [K]} r(X_t, a; \theta_t)$, where $\theta_t = (\theta_{t,a})_{a \in [K]}$
    Observe reward $Y_t \sim P(\cdot \mid X_t; \theta_{A_t})$ and update the posteriors.

---

## 3.2 Posterior approximation

The reward distribution is parameterized as a generalized linear model (GLM) [54], which allows for non-linear rewards. In addition, the diffusion model itself is highly non-linear due to the link functions $f_\ell$. These two sources of non-linearity make the posterior intractable, so we apply two layers of approximation: (i) a likelihood approximation to linearize the reward model, and (ii) a diffusion approximation to handle the non-linear hierarchy induced by the diffusion model prior.

**(i) Likelihood approximation.** We use an approach similar to the Laplace approximation, but instead of approximating the entire posterior, we approximate only the likelihood by a Gaussian. Precisely, the reward distribution $P(\cdot \mid x; \theta_a)$ belongs to the exponential family with mean function $g$. Thus

$$\prod_{i \in S_{t,a}} P(Y_i \mid X_i; \theta_a) \approx \mathcal{N}(\theta_a; \hat{B}_{t,a}, \hat{G}_{t,a}^{-1}), \tag{7}$$

where $\hat{B}_{t,a}$ is the maximum likelihood estimate and $\hat{G}_{t,a}$ is the Hessian of the negative log-likelihood:

$$\hat{B}_{t,a} = \arg\max_{\theta_a \in \mathbb{R}^d} \sum_{i \in S_{t,a}} \log P(Y_i \mid X_i; \theta_a), \qquad \hat{G}_{t,a} = \sum_{i \in S_{t,a}} \dot{g}(X_i^\top \hat{B}_{t,a}) X_i X_i^\top, \tag{8}$$

and $S_{t,a} = \{\ell \in [t-1] : A_\ell = a\}$ is the set of rounds in which action $a$ was selected. Unlike Laplace, which fits a global Gaussian to the full posterior, this step only linearizes the local likelihood, allowing the hierarchical diffusion structure of the prior to remain intact and expressive.

**(ii) Diffusion approximation.** Plugging the Gaussian likelihood approximation (7) into the posterior expressions $p(\theta_a \mid \psi_1, H_{t,a})$ and $p(\psi_{\ell-1} \mid \psi_\ell, H_t)$ removes the non-linearity of the reward model. However, the diffusion hierarchy remains non-linear through $f_\ell$. To handle this, we build on the closed-form posteriors of the *linear diffusion case* (where $f_\ell(\psi_\ell) = W_\ell \psi_\ell$; see Appendix B) and generalize them by replacing the linear terms $W_\ell \psi_\ell$ with their non-linear counterparts $f_\ell(\psi_\ell)$. This substitution yields a *posterior diffusion model* that retains the same hierarchical form as the prior but with data-dependent means and covariances. Details on how we transition from the linear to the general non-linear setting are provided in Appendices B and C. The resulting approximate posteriors (both action and latent) admit the following closed-form expressions.

**Approximate action posterior.** We approximate the conditional action posterior as

$$p(\theta_a \mid \psi_1, H_{t,a}) \approx \mathcal{N}(\theta_a; \hat{\mu}_{t,a}, \hat{\Sigma}_{t,a}),$$

where

$$\hat{\Sigma}_{t,a}^{-1} = \underbrace{\Sigma_1^{-1}}_{\text{prior precision}} + \underbrace{\hat{G}_{t,a}}_{\text{data precision}}, \qquad \hat{\mu}_{t,a} = \hat{\Sigma}_{t,a}\Big( \underbrace{\Sigma_1^{-1} f_1(\psi_1)}_{\text{prior contribution}} + \underbrace{\hat{G}_{t,a}\hat{B}_{t,a}}_{\text{data contribution}} \Big). \tag{9}$$

This posterior update has a clear interpretation. The posterior precision $\hat{\Sigma}_{t,a}^{-1}$ is the sum of the prior precision and the *data precision*. The posterior mean $\hat{\mu}_{t,a}$ is the precision-weighted average of the prior mean and the MLE $\hat{B}_{t,a}$. As more data are observed, the covariance shrinks and the mean moves from the prior mean $f_1(\psi_1)$ toward the MLE $\hat{B}_{t,a}$. When no data are available ($\hat{G}_{t,a} = 0$), the posterior reduces to the prior $\mathcal{N}(f_1(\psi_1), \Sigma_1)$; in the limit of infinite data ($\hat{G}_{t,a} \to \infty$), the posterior collapses to the MLE $\hat{B}_{t,a}$, with $\hat{\mu}_{t,a} \to \hat{B}_{t,a}$ and $\hat{\Sigma}_{t,a} \to 0$.

**Approximate latent posteriors.** For each $\ell \in [L+1] \setminus \{1\}$, we approximate the latent posterior as

$$p(\psi_{\ell-1} \mid \psi_\ell, H_t) \approx \mathcal{N}(\psi_{\ell-1}; \bar{\mu}_{t,\ell-1}, \bar{\Sigma}_{t,\ell-1}),$$

with

$$\bar{\Sigma}_{t,\ell-1}^{-1} = \underbrace{\Sigma_\ell^{-1}}_{\text{prior precision}} + \underbrace{\bar{G}_{t,\ell-1}}_{\text{data precision}} \,, \quad \bar{\mu}_{t,\ell-1} = \bar{\Sigma}_{t,\ell-1}\Big( \underbrace{\Sigma_\ell^{-1} f_\ell(\psi_\ell)}_{\text{prior contribution}} + \underbrace{\bar{B}_{t,\ell-1}}_{\text{data contribution}} \Big), \quad (10)$$

where, by convention, $f_{L+1}(\psi_{L+1}) = 0$ since the top layer $\psi_L$ has no parent $\psi_{L+1}$. The quantities $\bar{G}_{t,\ell}$ and $\bar{B}_{t,\ell}$ are computed recursively. The base recursion is

$$\bar{G}_{t,1} = \sum_{a=1}^{K} \left( \Sigma_1^{-1} - \Sigma_1^{-1}\hat{\Sigma}_{t,a}\Sigma_1^{-1} \right), \qquad \bar{B}_{t,1} = \Sigma_1^{-1} \sum_{a=1}^{K} \hat{\Sigma}_{t,a}\hat{G}_{t,a}\hat{B}_{t,a}, \quad (11)$$

and for each $\ell \in [L] \setminus \{1\}$,

$$\bar{G}_{t,\ell} = \Sigma_\ell^{-1} - \Sigma_\ell^{-1}\bar{\Sigma}_{t,\ell-1}\Sigma_\ell^{-1}, \qquad \bar{B}_{t,\ell} = \Sigma_\ell^{-1}\bar{\Sigma}_{t,\ell-1}\bar{B}_{t,\ell-1}. \quad (12)$$

The latent posterior update in Eq. (10) has the same structure as the action posterior. The posterior precision $\bar{\Sigma}_{t,\ell-1}^{-1}$ is the sum of the prior and data precisions , and the posterior mean is their precision-weighted combination. The data terms $\bar{G}_{t,\ell-1}$ and $\bar{B}_{t,\ell-1}$ are computed recursively (Eqs. (11) and (12)), so information collected at the action level propagates upward through the hierarchy.

**Interpretation.** The resulting approximate posterior remains a diffusion model whose conditional Gaussians have updated, data-dependent means and covariances. The latent-posterior means can be viewed as *refined link functions*:

$$\hat{f}_{t,\ell}(\psi_\ell) = \bar{\mu}_{t,\ell-1} = \bar{\Sigma}_{t,\ell-1}\left( \Sigma_\ell^{-1} f_\ell(\psi_\ell) + \bar{B}_{t,\ell-1} \right),$$

and $\bar{\Sigma}_{t,\ell}$ represents their updated uncertainty. Both are updated with data: covariances contract as uncertainty decreases, and means move from the prior toward the MLE. Unlike a full Laplace approximation, this formulation preserves the expressiveness of the posterior rather than replacing it globally with a single Gaussian, while also avoiding the heavy computation required by other approximate inference methods.

### 3.3 Extension to single shared action parameter

For the shared-parameter model in Remark 2.1, dTS's posterior approximations are similar. The action posterior is $p(\theta \mid \psi_1, H_t) \approx \mathcal{N}(\hat{\mu}_t, \hat{\Sigma}_t)$, where

$$\hat{\Sigma}_t^{-1} = \Sigma_1^{-1} + \hat{G}_t \,, \qquad \hat{\mu}_t = \hat{\Sigma}_t\left( \Sigma_1^{-1} f_1(\psi_1) + \hat{G}_t\hat{B}_t \right). \quad (13)$$

where

$$\hat{B}_t = \arg\max_{\theta \in \mathbb{R}^d} \sum_{i<t} \log P\left( Y_i \mid \varphi(X_i, A_i)^\top \theta \right), \quad \hat{G}_t = \sum_{i<t} \dot{g}\left( \varphi(X_i, A_i)^\top \hat{B}_t \right) \varphi(X_i, A_i)\varphi(X_i, A_i)^\top.$$

Similarly, for $\ell \in [L+1] \setminus \{1\}$, the latent posterior is $p(\psi_{\ell-1} \mid \psi_\ell, H_t) \approx \mathcal{N}(\bar{\mu}_{t,\ell-1}, \bar{\Sigma}_{t,\ell-1})$, where

$$\bar{\Sigma}_{t,\ell-1}^{-1} = \Sigma_\ell^{-1} + \bar{G}_{t,\ell-1}, \qquad \bar{\mu}_{t,\ell-1} = \bar{\Sigma}_{t,\ell-1}\left( \Sigma_\ell^{-1} f_\ell(\psi_\ell) + \bar{B}_{t,\ell-1} \right), \quad (14)$$

where, by convention, $f_{L+1}(\psi_{L+1}) = 0$ and the quantities $\bar{G}_{t,\ell}$ and $\bar{B}_{t,\ell}$ are computed recursively as

$$\text{Base case:} \qquad \bar{G}_{t,1} = \Sigma_1^{-1} - \Sigma_1^{-1}\hat{\Sigma}_t\Sigma_1^{-1}, \qquad \bar{B}_{t,1} = \Sigma_1^{-1}\hat{\Sigma}_t\hat{G}_t\hat{B}_t. \quad (15)$$

$$\text{Recursive case:} \qquad \bar{G}_{t,\ell} = \Sigma_\ell^{-1} - \Sigma_\ell^{-1}\bar{\Sigma}_{t,\ell-1}\Sigma_\ell^{-1}, \qquad \bar{B}_{t,\ell} = \Sigma_\ell^{-1}\bar{\Sigma}_{t,\ell-1}\bar{B}_{t,\ell-1}. \quad (16)$$

Again, this shared-parameter variant of dTS is presented for completeness and to illustrate the generality of our posterior derivations; the main focus of the paper remains on the per-action formulation in Eq. (1). Unless stated otherwise, all theoretical results and experiments use the main version of dTS described in Algorithm 1.

# 4 Informal theoretical insights

In this section, we present an informal Bayes regret analysis of `dTS` to build intuition around its statistical efficiency. We assume a simplified linear–Gaussian setting to make the analysis tractable: the reward distribution is linear-Gaussian and each link function $f_\ell(\psi_\ell) = W_\ell \psi_\ell$ is a known linear mixing matrix. These assumptions induce a hierarchy of $L$ linear Gaussian layers from the latent root to the action parameters. In this case, our posterior approximation becomes exact which enables an analysis reminiscent of linear contextual bandits [3]. However, our recursive hierarchical structure introduces technical differences: the posteriors must be derived inductively using total covariance decompositions, and regret bounds require tracking information flow across all latent layers. We emphasize that this regret bound does not hold in the general nonlinear case studied in experiments and on which we focus in this paper, and is only included here to provide theoretical intuition under simplifying assumptions. Formal statements and derivations are provided in Appendices E and F.

**Informal Bayes regret bound.** The bound of `dTS` in this case is

$$\tilde{\mathcal{O}}\left(\sqrt{n(dK\sigma_1^2 + d\sum_{\ell=1}^{L}\sigma_{\ell+1}^2\sigma_{\text{MAX}}^{2\ell})}\right),$$

where $\sigma_{\text{MAX}}^2 = \max_{\ell\in[L+1]} 1 + \frac{\sigma_\ell^2}{\sigma^2}$. This dependence on the horizon $n$ aligns with prior Bayes regret bounds scaling with $n$. However, the bound comprises $L + 1$ main terms. First, one relates to action parameters learning, conforming to a standard form [52], while the $L$ remaining terms are associated with learning each of the latent parameters.

**Sparsity refinement.** If each mixing matrix exhibits column sparsity, that, $W_\ell = (\bar{W}_\ell, 0_{d,d-d_\ell})$ with $d_\ell \ll d$ active columns, then the bound becomes

$$\mathcal{BR}(n) = \tilde{\mathcal{O}}\left(\sqrt{n(dK\sigma_1^2 + \sum_{\ell=1}^{L}d_\ell\sigma_{\ell+1}^2\sigma_{\text{MAX}}^{2\ell})}\right).$$

Hence, informative, *sparse* priors can cut the cost of learning deep latent chains down from $d$ to $d_\ell$. This Bayes regret bound has a clear interpretation: if the true environment parameters are drawn from the prior, then the expected regret of an algorithm stays below that bound. Consequently, a less informative prior (such as high variance) leads to a more challenging problem and thus a higher bound. Then, smaller values of $K$, $L$, $d$, $d_\ell$ translate to fewer parameters to learn, leading to lower regret. The regret also decreases when the initial variances $\sigma_\ell^2$ decrease. These dependencies are common in Bayesian analysis, and empirical results match them.

The reader might question the dependence of our bound on both $L$ and $K$. Details can be found in Appendix D.2, but in summary, we model the relationship between $\theta_a$ and $\psi_1$ stochastically as $\mathcal{N}(W_1\psi_1, \sigma_1^2 I_d)$ to account for potential nonlinearity. This choice makes the model robust to model misspecification but introduces extra uncertainty and requires learning both the $\theta_a$ and the $\psi_\ell$. This results in a regret bound that depends on both $K$ and $L$. However, thanks to the use of informative priors, our bound has significantly smaller constants compared to both the Bayesian regret for `LinTS` and its frequentist counterpart, as demonstrated empirically in Appendix G.5 where it is much tighter than both and in Section 4.1 where we theoretically compare our Bayes regret bound to that of `LinTS`.

**Technical contributions.** `dTS` uses hierarchical sampling. Thus the marginal posterior distribution of $\theta_a \mid H_t$ is not explicitly defined. The first contribution is deriving $\theta_a \mid H_t$ using the total covariance decomposition combined with an induction proof, as our posteriors were derived recursively. Unlike standard analyses where the posterior distribution of $\theta_a \mid H_t$ is predetermined due to the absence of latent parameters, our method necessitates this recursive total covariance decomposition. Moreover, in standard proofs, we need to quantify the increase in posterior precision for the action taken $A_t$ in each round $t \in [n]$. However, in `dTS`, our analysis extends beyond this. We not only quantify the posterior information gain for the taken action but also for every latent parameter, since they are also learned. To elaborate, we use our recursive posteriors that connect the posterior covariance of each latent parameter $\psi_\ell$ with the covariance of the posterior action parameters $\theta_a$. This allows us to propagate the information gain associated with the action taken in round $A_t$ to all latent parameters $\psi_\ell$, for $\ell \in [L]$ by induction. Details are given in Appendix F.

**Limitations.** We identified several limitations that should be addressed in future work. First, our Bayes regret analysis is established only for the *linear–Gaussian* case, where the diffusion prior collapses to a hierarchy of Gaussian distributions and `dTS` becomes exact Thompson Sampling. While this setting does not require a diffusion model, it validates our posterior approximation (exact in this

limit) and clarifies how prior structure and diffusion depth $L$ affect regret. Extending the theory to nonlinear diffusion or non-Gaussian rewards remains open. Second, for general nonlinear cases, `dTS` employs (i) a Laplace approximation for the reward likelihood and (ii) layer-wise linearization of diffusion links. A full analysis should account for errors coming from both approximations. We leave formal guarantees for future work. Third, `dTS` relies on a pre-trained diffusion prior. With scarce or biased offline data, the prior may be under-regularized. Empirically, performance degrades gracefully: `dTS` still outperforms LinTS and HierTS with as little as 1–5% pretraining data. Overall, `dTS` is advantageous when actions exhibit structured correlations and some offline data exist. In unstructured or purely online regimes, simpler methods such as `LinTS` may suffice.

## 4.1 Discussion

**Computational benefits.** Action correlations prompt an intuitive approach: marginalize all latent parameters and maintain a joint posterior of $(\theta_a)_{a \in [K]} \mid H_t$. Unfortunately, this is computationally inefficient for large action spaces. To illustrate, suppose that all posteriors are multivariate Gaussians. Then maintaining the joint posterior $(\theta_a)_{a \in [K]} \mid H_t$ necessitates converting and storing its $dK \times dK$-dimensional covariance matrix, leading to $\mathcal{O}(K^3 d^3)$ and $\mathcal{O}(K^2 d^2)$ time and space complexities. In contrast, the time and space complexities of `dTS` are $\mathcal{O}((L+K)d^3)$ and $\mathcal{O}((L+K)d^2)$. This is because `dTS` requires converting and storing $L + K$ covariance matrices, each being $d \times d$-dimensional. The improvement is huge when $K \gg L$, which is common in practice. Certainly, a more straightforward way to enhance computational efficiency is to discard latent parameters and maintain $K$ individual posteriors, each relating to an action parameter $\theta_a \in \mathbb{R}^d$ (`LinTS`). This improves time and space complexity to $\mathcal{O}(Kd^3)$ and $\mathcal{O}(Kd^2)$. However, `LinTS` maintains independent posteriors and fails to capture the correlations among actions; it only models $\theta_a \mid H_{t,a}$ rather than $\theta_a \mid H_t$ as done by `dTS`. Consequently, `LinTS` incurs higher regret due to the information loss caused by unused interactions of similar actions. Our regret bound and empirical results reflect this aspect.

**Statistical benefits.** We do not provide a matching lower bound. The only Bayesian lower bound that we know of is $\Omega(\log^2(n))$ for a much simpler $K$-armed bandit [45, Theorem 3]. All seminal works on Bayesian bandits do not match it and providing such lower bounds on Bayes regret is still relatively unexplored (even in standard settings) compared to the frequentist one. Also, a min-max lower bound of $\Omega(d\sqrt{n})$ was given by Dani et al. [21]. In this work, we argue that our bound reflects the overall structure of the problem by comparing `dTS` to algorithms that only partially use the structure or do not use it at all as follows.

When the link functions are linear, we can transform the diffusion prior into a Bayesian linear model (`LinTS`) by marginalizing out the latent parameters; in which case the prior on action parameters becomes $\theta_a \sim \mathcal{N}(0, \Sigma)$, with the $\theta_a$ being not necessarily independent, and $\Sigma$ is the marginal initial covariance of action parameters and it writes $\Sigma = \sigma_1^2 I_d + \sum_{\ell=1}^{L} \sigma_{\ell+1}^2 B_\ell B_\ell^\top$ with $B_\ell = \prod_{i=1}^{\ell} W_i$. Then, it is tempting to directly apply `LinTS` to solve our problem. This approach will induce higher regret because the additional uncertainty of the latent parameters is accounted for in $\Sigma$ despite integrating them. This causes the *marginal* action uncertainty $\Sigma$ to be much higher than the *conditional* action uncertainty $\sigma_1^2 I_d$, since we have $\Sigma = \sigma_1^2 I_d + \sum_{\ell=1}^{L} \sigma_{\ell+1}^2 B_\ell B_\ell^\top \succcurlyeq \sigma_1^2 I_d$. This discrepancy leads to higher regret, especially when $K$ is large. This is due to `LinTS` needing to learn $K$ independent $d$-dimensional parameters, each with a considerably higher initial covariance $\Sigma$. This is also reflected by our regret bound. To simply comparisons, suppose that $\sigma \geq \max_{\ell \in [L+1]} \sigma_\ell$ so that $\sigma_{\text{MAX}}^2 \leq 2$. Then the regret bounds of `dTS` (where we bound $\sigma_{\text{MAX}}^{2\ell}$ by $2^\ell$) and `LinTS` read

$$\texttt{dTS} : \tilde{\mathcal{O}}\left(\sqrt{n(dK\sigma_1^2 + \sum_{\ell=1}^{L} d_\ell \sigma_{\ell+1}^2 2^\ell)}\right), \qquad \texttt{LinTS} : \tilde{\mathcal{O}}\left(\sqrt{ndK(\sigma_1^2 + \sum_{\ell=1}^{L} \sigma_{\ell+1}^2)}\right).$$

Then regret improvements are captured by the variances $\sigma_\ell$ and the sparsity dimensions $d_\ell$, and we proceed to illustrate this through the following scenarios.

**(I) Decreasing variances.** Assume that $\sigma_\ell = 2^\ell$ for any $\ell \in [L + 1]$. Then, the regrets become

$$\texttt{dTS} : \tilde{\mathcal{O}}\left(\sqrt{n(dK + \sum_{\ell=1}^{L} d_\ell 4^\ell))}\right), \qquad \texttt{LinTS} : \tilde{\mathcal{O}}\left(\sqrt{ndK2^L}\right)$$

Now to see the order of gain, assume the problem is high-dimensional ($d \gg 1$), and set $L = \log_2(d)$ and $d_\ell = \lfloor \frac{d}{2^\ell} \rfloor$. Then the regret of `dTS` becomes $\tilde{\mathcal{O}}\left(\sqrt{nd(K + L)}\right)$, and hence the multiplicative factor $2^L$ in `LinTS` is removed and replaced with a smaller additive factor $L$.

**(II) Constant variances.** Assume that $\sigma_\ell = 1$ for any $\ell \in [L+1]$. Then, the regrets become

$$\texttt{dTS} : \tilde{\mathcal{O}}\big(\sqrt{n(dK + \sum_{\ell=1}^{L} d_\ell 2^\ell))}\big), \qquad \texttt{LinTS} : \tilde{\mathcal{O}}\big(\sqrt{ndKL}\big)$$

Similarly, let $L = \log_2(d)$, and $d_\ell = \lfloor \frac{d}{2^\ell} \rfloor$. Then dTS's regret is $\tilde{\mathcal{O}}\big(\sqrt{nd(K+L)}\big)$. Thus the multiplicative factor $L$ in LinTS is removed and replaced with the additive factor $L$. By comparing this to **(I)**, the gain with decreasing variances is greater than with constant ones. In general, diffusion models use decreasing variances [30] and hence we expect great gains in practice. All observed improvements in this section could become even more pronounced when employing non-linear diffusion models. In our theory, we used linear diffusion models, and yet we can already discern substantial differences. Moreover, under non-linear diffusion Eq. (1), the latent parameters cannot be analytically marginalized, making LinTS with exact marginalization inapplicable.

**Regret independent of $K$?** dTS's regret bound scales with $K\sigma_1^2$ rather than $K\sum_\ell \sigma_\ell^2$, which is particularly advantageous when $\sigma_1$ is small, as is often the case with diffusion model priors. Both our theoretical bound and empirical results show that dTS's advantage over LinTS increases as the action space grows. Nevertheless, dTS's regret still depends on $K$; this dependence arises from the problem setting rather than from the algorithm itself. Prior works [25, 67, 70] have proposed bandit algorithms whose regret does not scale with $K$. This difference stems from the setting considered: we study the *disjoint* (per-action) case $r(x, a; \theta) = x^\top \theta_a$, where $\theta = (\theta_a)_{a \in [K]} \in \mathbb{R}^{dK}$, requiring the learning of $Kd$ parameters and thus introducing an inherent dependence on $K$ when $\sigma_1 > 0$. In contrast, $K$-independent regret results are obtained in the *shared-parameter* setting described in Remark 2.1, where $r(x, a; \theta) = \varphi(x, a)^\top \theta$ and only a single $d$-dimensional parameter must be learned. However, this formulation requires access to the feature map $\varphi$. Fortunately, dTS is also compatible with this setting (Section 3.3), in which case its regret would indeed be independent of $K$.

## 5 Experiments

**Experimental setup.** We evaluate dTS using both synthetic and MovieLens problems. In our experiments, we run 50 random simulations and plot the average regret with standard error. Our main contribution is to demonstrate that pretraining a diffusion model offline enables the construction of expressive and informative priors that substantially improve exploration efficiency in contextual bandits. We first evaluate dTS in a setting where the prior matches the true generative process (Section 5.1 to isolate the benefit of informative priors), and then consider a misspecified regime (Section 5.2 and Appendix G) where the prior is either trained on out-of-distribution data or intentionally perturbed. These experiments show that even when the prior is imperfect, dTS maintains strong performance: highlighting its robustness and practical relevance. Code can be found in this GitHub repository.

### 5.1 True prior is a diffusion model

Synthetic bandit problems are generated from the diffusion model in Eq. (1) with both linear and non-linear rewards. Linear rewards follow $P(\cdot \mid x; \theta_a) = \mathcal{N}(x^\top \theta_a, 1)$, while non-linear rewards are binary from $P(\cdot \mid x; \theta_a) = \text{Ber}(g(x^\top \theta_a))$, with $g$ as the sigmoid function. Covariances are $\Sigma_\ell = I_d$, and contexts $X_t$ are uniformly drawn from $[-1, 1]^d$. We vary $d \in \{5, 20\}$, $L \in \{2, 4\}$, $K \in \{10^2, 10^4\}$, and set the horizon to $n = 5000$, considering both linear and non-linear models.

**Linear diffusion.** We consider Eq. (1) with $f_\ell(\psi) = W_\ell \psi$, where $W_\ell$ uniformly drawn from $[-1, 1]^{d \times d}$. Sparsity is introduced by zeroing the last $d_\ell$ columns of $W_\ell$ as $W_\ell = (\bar{W}_\ell, 0_{d, d-d_\ell})$. For $d = 5$ and $L = 2$, $(d_1, d_2) = (5, 2)$; for $d = 20$ and $L = 4$, $(d_1, d_2, d_3, d_4) = (20, 10, 5, 2)$.

**Non-linear diffusion.** We consider Eq. (1) where $f_\ell$ are 2-layer neural networks with random weights in $[-1, 1]$, ReLU activation, and hidden layers of size $h = 20$ for $d = 5$, and $h = 60$ for $d = 20$.

**Baselines.** For linear rewards, we use LinUCB [1], LinTS [3], and HierTS [33], marginalizing out all latent parameters except $\psi_L$, which corresponds to HierTS-1 in Appendix D.1. For non-linear rewards, we include UCB-GLM [50] and GLM-TS [18]. We exclude GLM-UCB [24] due to high regret and HierTS as it's designed for linear rewards. We name dTS as dTS-dr, where d refers to diffusion type (L for linear, N for non-linear) and r indicates reward type (L for linear, N for non-linear). For example, dTS-LL signifies dTS in linear diffusion with linear rewards.

**Results and interpretations.** Results are shown in Fig. 1 and we make the following observations:

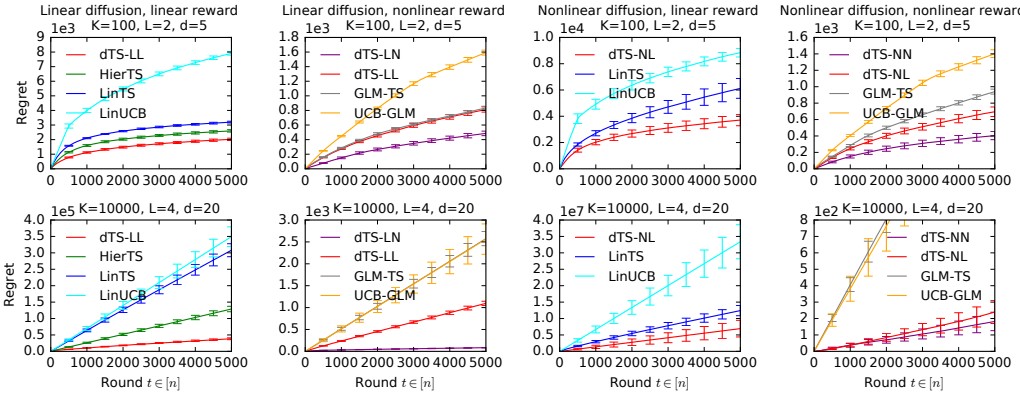

Figure 1: Regret of dTS with varying diffusion and reward models and varying parameters $d$, $K$, $L$.

**1) dTS demonstrates superior performance (Fig. 1).** dTS consistently outperforms the baselines across all settings, including the four combinations of linear/non-linear diffusion and reward (columns in Fig. 1) and both bandit settings with varying $K$, $L$, and $d$ (rows in Fig. 1).

**2) Latent diffusion structure may be more important than the reward distribution.** When rewards are non-linear (second and fourth columns in Fig. 1), we include variants of dTS that use the correct diffusion prior but the wrong reward distribution, applying linear-Gaussian instead of logistic-Bernoulli (dTS-LL in the second column and dTS-NL in the fourth). Despite the reward misspecification, these variants outperform models using the correct reward distribution but ignoring the latent diffusion structure, such as GLM-TS and UCB-GLM. This highlights the importance of accounting for latent structure, which can be more critical than an accurate reward distribution.

**3) Performance gap between dTS and LinTS widens as $K$ increases (Fig. 2a).** To show dTS's improved scalability, we evaluate its performance with varying values of $K \in [10, 5 \times 10^4]$, in the linear diffusion and rewards setting. Fig. 2a shows the final cumulative regret for varying $K$ values for both dTS-LL and LinTS, revealing a widening performance gap as $K$ increases.

**4) Regret scaling with $K$, $d$ and $L$ matches our theory (Fig. 2b).** We assess the effect of the number of actions $K$, context dimension $d$, and diffusion depth $L$ on dTS's regret. Using the linear diffusion and rewards setting, for which we have derived a Bayes regret upper bound, we plot dTS-LL's regret across varying values of $K \in \{10, 100, 500, 1000\}$, $d \in \{5, 10, 15, 20\}$, and $L \in \{2, 4, 5, 6\}$ in Fig. 2b. As predicted by our theory, the empirical regret increases with larger values of $K$, $d$, or $L$, as these make the learning problem more challenging, leading to higher regret.

**5) Diffusion prior misspecification (Fig. 2c).** Here, dTS's diffusion prior parameters differ from the true diffusion prior. In the linear diffusion and reward setting, we replace the true parameters $W_\ell$ and $\Sigma_\ell$ with misspecified ones, $W_\ell + \epsilon_1$ and $\Sigma_\ell + \epsilon_2$, where $\epsilon_1$ and $\epsilon_2$ are uniformly sampled from $[v, v + 0.5]^{d \times d}$, with $v$ controlling the misspecification level. We vary $v \in \{0.5, 1, 1.5\}$ and assess dTS's performance, comparing it to the well-specified dTS-LL and the strongest baseline in this fully-linear setting, HierTS. As shown in Fig. 2c, dTS's performance decreases with increasing misspecification but remains superior to the baseline, except at $v = 1.5$, where their performances are comparable. Additional misspecification experiments are presented in Section 5.2, where the bandit environment is not sampled from a diffusion model.

## 5.2 True prior is not a diffusion model

**Swiss roll data.** Unlike previous experiments, the true action parameters are now sampled from the Swiss roll distribution (see Fig. 4 in Appendix G.1), rather than from a diffusion model. The diffusion model used by dTS is pre-trained on samples from this distribution, with the offline pre-training procedure described in Appendix G.2. Fig. 3a shows that larger sample sizes increase the performance gap between dTS and LinTS. More samples improve the estimation of the diffusion prior (see Fig. 4 in Appendix G.1), leading to better dTS performance. Notably, comparable performance was achieved with as few as 10 samples, and dTS outperformed LinTS by a factor of 1.5 with just 50 samples.

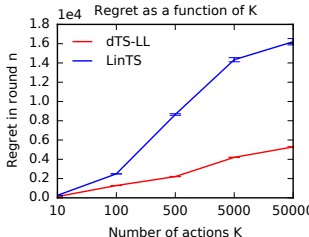
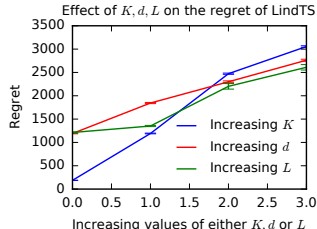
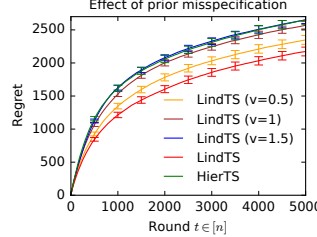

(a) Perf. gap increases with $K$.    (b) Regret scaling with $K, d, L$.    (c) Diffusion prior misspecification.

Figure 2: Effect of various factors on dTS's performance.

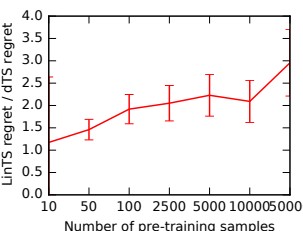
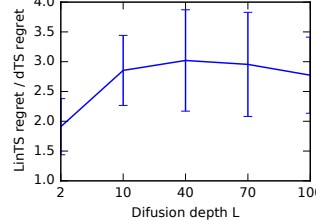
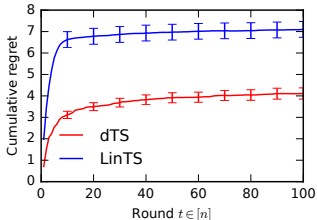

(a) Ratio of LinTS/dTS cumulative regret in the last round with varying pre-training sample size in $[10, 5 \times 10^4]$. **Higher values mean a bigger performance gap**.

(b) Ratio of LinTS/dTS cumulative regret in the last round with varying diffusion depth $L$ in $[2, 100]$. **Higher values mean a bigger performance gap**.

(c) Regret of dTS in **MovieLens**. The diffusion model with $L = 40$ is pre-trained on embeddings obtained by low-rank factorization of Movie-Lens rating matrix.

Figure 3: **(a) and (b):** Impact of pre-training sample size and diffusion depth $L$ for the **Swiss roll data**. **(c):** Regret of dTS in **MovieLens**.

While more samples may be required for more complex problems, LinTS would also struggle in such cases. Therefore, we expect these gains to be even more significant in more challenging settings.

We studied the effect of the pre-trained diffusion model depth $L$ and found that $L \approx 40$ yields the best performance, with a drop beyond that point (Fig. 3b). While our theory doesn't apply directly here, as it assumes a linear diffusion model, it still offers some intuition on the decreased performance for $L > 40$. The theorem shows dTS's regret bound increases with $L$ when the true distribution is a diffusion model. For small $L$, the pre-trained model doesn't fully capture the true distribution, making the theorem inapplicable, but at $L \approx 40$, the distribution is nearly captured, and further increases in $L$ lead to higher regret, consistent with our theory.

**MovieLens data.** We also evaluate dTS using the standard MovieLens [46] setting. In this semi-synthetic experiment, a user is sampled from the rating matrix in each interaction round, and the reward is the rating the user gives to a movie (see Clavier et al. [20, Section 5] for details about this setting). Here, the true distribution of action parameters is unknown and not a diffusion model. The diffusion model is pre-trained on offline estimates of action parameters obtained through low-rank factorization of the rating matrix. Fig. 3c demonstrates that dTS outperforms LinTS in this setting. Additional **CIFAR-10** ablation studies are provided in Appendix G.4 where similar strong improvements are observed .

## 6   Conclusion

We use a pre-trained diffusion model as a strong and flexible prior for dTS. Diffusion pre-training leverages abundant offline data, which is then fine-tuned through online interactions via our tractable posterior approximation. This approximation enables efficient posterior sampling and updates while maintaining strong empirical performance. Moreover, dTS admits a simple Bayesian regret bound in the linear–Gaussian setting. Broader impact and computational considerations are discussed in Appendices I and J, and directions for future work are provided in Appendix H.

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

## Supplementary materials

**Notation.** For any positive integer $n$, we define $[n] = \{1, 2, ..., n\}$. Let $v_1, \ldots, v_n \in \mathbb{R}^d$ be $n$ vectors, $(v_i)_{i \in [n]} \in \mathbb{R}^{nd}$ is the $nd$-dimensional vector obtained by concatenating $v_1, \ldots, v_n$. For any matrix $A \in \mathbb{R}^{d \times d}$, $\lambda_1(A)$ and $\lambda_d(A)$ denote the maximum and minimum eigenvalues of A, respectively. Finally, we write $\tilde{\mathcal{O}}$ for the big-O notation up to polylogarithmic factors.

**Table of notations.**

Table 1: Notation.

| Symbol | Definition |
|---|---|
| $n$ | Learning horizon |
| $\mathcal{X}$ | Context space |
| $K$ | Number of actions |
| $[K]$ | Set of actions |
| $d$ | Dimension of contexts and action parameters $d$ |
| $\theta_a$ | $d$-dimensional parameter of action $a \in [K]$ |
| $P(\cdot \mid x; \theta_a)$ | Reward distribution of context $x$ and action $a$ |
| $r(x, a; \theta_*)$ | Reward function of context $x$ and action $a$ |
| $\mathcal{BR}(n)$ | Bayes regret after $n$ interactions |
| $\mathcal{N}(\mu, \Sigma)$ | Multivariate Gaussian distribution of parameters $\mu$ and $\Sigma$ |
| $\mathcal{N}(\cdot; \mu, \Sigma)$ | Multivariate Gaussian density of parameters $\mu$ and $\Sigma$ |
| $L$ | Diffusion model depth |
| $\psi_\ell$ | $\ell$-th $d$-dimensional latent parameter |
| $f_\ell$ | Link functions of the diffusion model |
| $\Sigma_\ell$ | Covariances of the link function |
| $H_t$ | History of interactions |

## A   Extended related work

**Thompson sampling (TS)** operates within the Bayesian framework and it involves specifying a prior/likelihood model. In each round, the agent samples unknown model parameters from the current posterior distribution. The chosen action is the one that maximizes the resulting reward. TS is naturally randomized, particularly simple to implement, and has highly competitive empirical performance in both simulated and real-world problems [59, 18]. Regret guarantees for the TS heuristic remained open for decades even for simple models. Recently, however, significant progress has been made. For standard multi-armed bandits, TS is optimal in the Beta-Bernoulli model [37, 4], Gaussian-Gaussian model [4], and in the exponential family using Jeffrey's prior [39]. For linear bandits, TS is nearly-optimal [59, 5, 2]. In this work, we build TS upon complex diffusion priors and analyze the resulting Bayes regret [59, 55, 28] in the linear contextual bandit setting.

**Decision-making with diffusion models** gained attention recently, especially in offline learning [6, 36, 65]. However, their application in online learning was only examined by Hsieh et al. [34], which focused on meta-learning in multi-armed bandits without theoretical guarantees. In this work, we expand the scope of Hsieh et al. [34] to encompass the broader contextual bandit framework. In particular, we provide theoretical analysis for linear instances, effectively capturing the advantages of using diffusion models as priors in contextual Thompson sampling. These linear cases are particularly captivating due to closed-form posteriors, enabling both theoretical analysis and computational efficiency; an important practical consideration.

**Hierarchical Bayesian bandits** [13, 43, 14, 61, 63, 33, 56, 64, 8] applied TS to simple graphical models, wherein action parameters are generally sampled from a Gaussian distribution centered at a single latent parameter. These works mostly span meta- and multi-task learning for multi-armed bandits, except in cases such as Aouali et al. [8], Hong et al. [32] that consider the contextual bandit setting. Precisely, Aouali et al. [8] assume that action parameters are sampled from a Gaussian distribution centered at a linear mixture of multiple latent parameters. On the other hand, Hong et al. [32] applied TS to a graphical model represented by a tree. Our work can be seen as an extension of all these works to much more complex graphical models, for which both theoretical and algorithmic

foundations are developed. Note that the settings in most of these works can be recovered with specific choices of the diffusion depth $L$ and functions $f_\ell$. This attests to the modeling power of dTS.

**Approximate Thompson sampling** is a major problem in the Bayesian inference literature. This is because most posterior distributions are intractable, and thus practitioners must resort to sophisticated computational techniques such as Markov chain Monte Carlo [41]. Prior works [57, 18, 42] highlight the favorable empirical performance of approximate Thompson sampling. Particularly, [42] provide theoretical guarantees for Thompson sampling when using the Laplace approximation in generalized linear bandits (GLB). In our context, we incorporate approximate sampling when the reward exhibits non-linearity. While our approximation does not come with formal guarantees, it enjoys strong practical performance. An in-depth analysis of this approximation is left as a direction for future works. Similarly, approximating the posterior distribution when the diffusion model is non-linear as well as analyzing it is an interesting direction of future works.

**Bandits with underlying structure** also align with our work, where we assume a structured relationship among actions, captured by a diffusion model. In latent bandits [53, 31], a single latent variable indexes multiple candidate models. Within structured finite-armed bandits [47, 29], each action is linked to a known mean function parameterized by a common latent parameter. This latent parameter is learned. TS was also applied to complex structures [69, 27]. However, simultaneous computational and statistical efficiencies aren't guaranteed. Meta- and multi-task learning with upper confidence bound (UCB) approaches have a long history in bandits [12, 26, 22, 16]. These, however, often adopt a frequentist perspective, analyze a stronger form of regret, and sometimes result in conservative algorithms. In contrast, our approach is Bayesian, with analysis centered on Bayes regret. Remarkably, our algorithm, dTS, performs well as analyzed without necessitating additional tuning. Finally, **Low-rank bandits** [35, 17, 68] also relate to our linear diffusion model when $L = 1$. Broadly, there exist two key distinctions between these prior works and the special case of our model (linear diffusion model with $L = 1$). First, they assume $\theta_a = W_1 \psi_1$, whereas we incorporate additional uncertainty in the covariance $\Sigma_1$ to account for possible misspecification as $\theta_a = \mathcal{N}(W_1 \psi_1, \Sigma_1)$. Consequently, these algorithms might suffer linear regret due to model misalignment. Second, we assume that the mixing matrix $W_1$ is available and pre-learned offline, whereas they learn it online. While this is more general, it leads to computationally expensive methods that are difficult to employ in a real-world online setting.

**Large action spaces.** The regret bound of dTS scales with $K\sigma_1^2$ rather than $K \sum_\ell \sigma_\ell^2$, which is particularly advantageous when $\sigma_1$ is small: a common case in diffusion models with decreasing layer variances. In the limiting case $\sigma_1 = 0$, the regret becomes independent of $K$. Our theoretical analysis (Section 4.1) and empirical results both show that the performance gap between dTS and LinTS widens as the number of actions increases, highlighting dTS's suitability for large action spaces.

Some prior works [25, 67, 70] achieve regret bounds that do not scale with $K$. This discrepancy stems from the problem setting rather than the algorithm itself. Specifically, those works adopt a *shared-parameter* model $r(x, a) = \varphi(x, a)^\top \theta$ with a single parameter $\theta \in \mathbb{R}^d$ and a known feature map $\varphi$, whereas we study the *disjoint* case $r(x, a) = x^\top \theta_a$ with $K$ separate $d$-dimensional parameters. In the shared-parameter setting (see Remark 2.1 and Section 3.3), dTS would similarly achieve regret independent of $K$.

In summary, the dependence on $K$ arises from the modeling choice rather than a limitation of dTS. When $\varphi$ is available, dTS scales only with $d$; otherwise, in the per-action setting, it remains both computationally and statistically efficient (Section 4.1). Empirically, even for very large action spaces (e.g., $K = 10^4$), dTS substantially outperforms existing baselines, with the performance gap increasing as $K$ grows: highlighting its scalability to large action spaces.

# B  Posterior derivations for linear diffusion models

Our posterior approximation builds on the simplified setting where the diffusion model is fully linear, i.e., each link function $f_\ell$ is linear in $\psi_\ell$. This linear case, studied in our earlier work [7], serves as the analytical foundation for our posterior approximation used in the general non-linear case. In Appendix C, we show how the exact posteriors derived in this linear setting inspire our efficient approximation, which extends naturally to practical diffusion models that are typically highly non-linear.

## B.1 Linear diffusion model

Here, we assume the link functions $f_\ell$ are linear such as $f_\ell(\psi_\ell) = W_\ell \psi_\ell$ for $\ell \in [L]$, where $W_\ell \in \mathbb{R}^{d \times d}$ are *known mixing matrices*. Then, Eq. (1) becomes a linear Gaussian system (LGS) [15] and can be summarized as follows

$$
\begin{aligned}
\psi_L &\sim \mathcal{N}(0, \Sigma_{L+1}), & & (17) \\
\psi_{\ell-1} \mid \psi_\ell &\sim \mathcal{N}(W_\ell \psi_\ell, \Sigma_\ell), & \forall \ell &\in [L]/\{1\}, \\
\theta_a \mid \psi_1 &\sim \mathcal{N}(W_1 \psi_1, \Sigma_1), & \forall a &\in [K], \\
Y_t \mid X_t, \theta_{A_t} &\sim P(\cdot \mid X_t; \theta_{A_t}), & \forall t &\in [n].
\end{aligned}
$$

This model is important, both in theory and practice. For theory, it leads to closed-form posteriors when the reward distribution is linear-Gaussian as $P(\cdot \mid x; \theta_a) = \mathcal{N}(\cdot; x^\top \theta_a, \sigma^2)$. This allows bounding the Bayes regret of dTS. For practice, the posterior expressions are used to motivate efficient approximations for the general case in Eq. (1) as we show in Section 3.2. These derivations can be proven following standard techniques [15], and the reader may refer to Aouali [7, Appendix B] for an example of how these posteriors can be derived in the case of contextual bandits.

## B.2 Posterior derivation in the linear diffusion case

We now consider the linear link function case, where $f_\ell(\psi_\ell) = W_\ell \psi_\ell$ for $\ell \in [L]$ (the setting above in Appendix B.1). Recall that the reward distribution is modeled as a generalized linear model (GLM) [54], allowing for non-linear rewards even when the diffusion links are linear. This non-linearity in the reward distribution prevents closed-form posteriors. However, since the non-linearity arises only through the reward likelihood, we approximate it by a Gaussian, leading to efficient posterior updates that are exact whenever the reward model itself is Gaussian; a special case of the GLM framework.

Specifically, let $P(\cdot \mid x; \theta_a)$ be an exponential-family distribution. The log-likelihood of the data associated with action $a$ is

$$
\log p(H_{t,a} \mid \theta_a) = \sum_{i \in S_{t,a}} \left[ Y_i X_i^\top \theta_a - A(X_i^\top \theta_a) + C(Y_i) \right],
$$

where $C$ is a real-valued function and $A$ is twice continuously differentiable, with derivative $\dot{A} = g$ representing the mean function. Let $\hat{B}_{t,a}$ and $\hat{G}_{t,a}$ denote the maximum likelihood estimate (MLE) and the Hessian of the negative log-likelihood, respectively:

$$
\hat{B}_{t,a} = \arg\max_{\theta_a \in \mathbb{R}^d} \log p(H_{t,a} \mid \theta_a), \qquad \hat{G}_{t,a} = \sum_{i \in S_{t,a}} \dot{g}(X_i^\top \hat{B}_{t,a}) X_i X_i^\top, \qquad (18)
$$

where $S_{t,a} = \{\ell \in [t-1] : A_\ell = a\}$ is the set of rounds in which action $a$ was taken up to round $t$. We approximate the likelihood as

$$
p(H_{t,a} \mid \theta_a) \approx \mathcal{N}(\theta_a; \hat{B}_{t,a}, \hat{G}_{t,a}^{-1}), \qquad (19)
$$

which renders all subsequent posteriors Gaussian. Once this approximation is done, all other derivations of the action posterior and latent posteriors are exact.

**Action posterior.** The conditional action posterior becomes

$$
p(\theta_a \mid \psi_1, H_{t,a}) \approx \mathcal{N}(\theta_a; \hat{\mu}_{t,a}, \hat{\Sigma}_{t,a}),
$$

with parameters

$$
\hat{\Sigma}_{t,a}^{-1} = \Sigma_1^{-1} + \hat{G}_{t,a}, \qquad \hat{\mu}_{t,a} = \hat{\Sigma}_{t,a} \left( \Sigma_1^{-1} W_1 \psi_1 + \hat{G}_{t,a} \hat{B}_{t,a} \right). \qquad (20)
$$

**Latent posteriors.** For each $\ell \in [L] \setminus \{1\}$, the conditional latent posterior is

$$
p(\psi_{\ell-1} \mid \psi_\ell, H_t) \approx \mathcal{N}(\psi_{\ell-1}; \bar{\mu}_{t,\ell-1}, \bar{\Sigma}_{t,\ell-1}),
$$

where

$$
\bar{\Sigma}_{t,\ell-1}^{-1} = \Sigma_\ell^{-1} + \bar{G}_{t,\ell-1}, \qquad \bar{\mu}_{t,\ell-1} = \bar{\Sigma}_{t,\ell-1} \left( \Sigma_\ell^{-1} W_\ell \psi_\ell + \bar{B}_{t,\ell-1} \right). \qquad (21)
$$

The top-layer posterior is

$$p(\psi_L \mid H_t) \approx \mathcal{N}(\psi_L; \bar{\mu}_{t,L}, \bar{\Sigma}_{t,L}),$$

with

$$\bar{\Sigma}_{t,L}^{-1} = \Sigma_{L+1}^{-1} + \bar{G}_{t,L}, \qquad\qquad \bar{\mu}_{t,L} = \bar{\Sigma}_{t,L} \bar{B}_{t,L}. \qquad (22)$$

**Recursive updates.** The matrices $\bar{G}_{t,\ell}$ and $\bar{B}_{t,\ell}$ for $\ell \in [L]$ are defined recursively. The base recursion is

$$\bar{G}_{t,1} = \mathrm{W}_1^\top \sum_{a=1}^K \left( \Sigma_1^{-1} - \Sigma_1^{-1} \hat{\Sigma}_{t,a} \Sigma_1^{-1} \right) \mathrm{W}_1, \qquad \bar{B}_{t,1} = \mathrm{W}_1^\top \Sigma_1^{-1} \sum_{a=1}^K \hat{\Sigma}_{t,a} \hat{G}_{t,a} \hat{B}_{t,a}. \qquad (23)$$

Then, for $\ell \in [L] \setminus \{1\}$, the recursive step is

$$\bar{G}_{t,\ell} = \mathrm{W}_\ell^\top \left( \Sigma_\ell^{-1} - \Sigma_\ell^{-1} \bar{\Sigma}_{t,\ell-1} \Sigma_\ell^{-1} \right) \mathrm{W}_\ell, \qquad \bar{B}_{t,\ell} = \mathrm{W}_\ell^\top \Sigma_\ell^{-1} \bar{\Sigma}_{t,\ell-1} \bar{B}_{t,\ell-1}. \qquad (24)$$

**Discussion.** This completes the derivation of the linear posterior approximation. All posteriors are Gaussian and exact whenever the reward distribution follows a linear-Gaussian model, i.e.

$$P(\cdot \mid x; \theta_a) = \mathcal{N}(\cdot; x^\top \theta_a, \sigma^2).$$

In this case, the above posterior updates coincide with the exact Bayesian updates, while for general GLMs they serve as efficient and accurate approximations.

## C  Posterior derivations for non-linear diffusion models

The general diffusion model (Eq. (1), which is our case of interest) involves two sources of non-linearity: (i) the reward distribution $P(\cdot \mid x; \theta)$, which may follow a non-linear generalized linear model (GLM), and (ii) the diffusion links $f_\ell(\psi_\ell)$, which can be arbitrary non-linear functions. Both sources make the posterior intractable, and therefore two approximations are needed.

**First approximation (likelihood).** We first approximate the reward likelihood by a Gaussian density (as we did above in Eq. (19)). After this substitution, the model becomes conditionally Gaussian given the latent variables. This step is exact when the reward model is linear-Gaussian, and approximate otherwise.

**Second approximation (diffusion hierarchy).** Even after the likelihood is approximated, the diffusion hierarchy remains non-linear because of the non-linear mappings $f_\ell$. To handle this, we reuse the exact Gaussian posteriors derived for the linear diffusion case (Appendix B.2) and generalize them as follows:

- Replace each linear mapping $\mathrm{W}_\ell \psi_\ell$ by its non-linear counterpart $f_\ell(\psi_\ell)$, which represents the mean of the diffusion prior at layer $\ell$.
- Remove matrix multiplications involving $\mathrm{W}_\ell$ in the recursive updates.

This step can be viewed as extending the linear-Gaussian posterior formulas to a general non-linear setting, without performing explicit linearization or optimization.

**Resulting approximation.** The two steps above yield a posterior where each conditional factor $p(\theta_a \mid \psi_1, H_{t,a})$ and $p(\psi_{\ell-1} \mid \psi_\ell, H_t)$ remains Gaussian with updated means and covariances, while the overall model retains the hierarchical diffusion structure. The approximation satisfies two desirable properties: it exactly recovers the diffusion prior when no data is available, and as more data is observed, the likelihood terms dominate and the prior influence fades naturally.

This approach is computationally efficient, avoids costly posterior sampling or variational optimization, and remains expressive since the overall posterior is still a diffusion model with non-linear link functions and covariances that are now data-dependent.

# D   Additional discussions

## D.1   Additional discussion: link to two-level hierarchies

The linear diffusion Eq. (17) can be marginalized into a 2-level hierarchy using two different strategies. The first one yields,

$$\psi_L \sim \mathcal{N}(0, \sigma_{L+1}^2 \mathrm{B}_L \mathrm{B}_L^\top), \tag{25}$$
$$\theta_a \mid \psi_L \sim \mathcal{N}(\psi_L, \Omega_1), \qquad\qquad \forall a \in [K],$$

with $\Omega_1 = \sigma_1^2 I_d + \sum_{\ell=1}^{L-1} \sigma_{\ell+1}^2 \mathrm{B}_\ell \mathrm{B}_\ell^\top$ and $\mathrm{B}_\ell = \prod_{i=1}^\ell \mathrm{W}_i$. The second strategy yields,

$$\psi_1 \sim \mathcal{N}(0, \Omega_2), \tag{26}$$
$$\theta_a \mid \psi_1 \sim \mathcal{N}(\psi_1, \sigma_1^2 I_d), \qquad\qquad \forall a \in [K],$$

where $\Omega_2 = \sum_{\ell=1}^{L} \sigma_{\ell+1}^2 \mathrm{B}_\ell \mathrm{B}_\ell^\top$. Recently, `HierTS` [33] was developed for such two-level graphical models, and we call `HierTS` under Eq. (25) by `HierTS-1` and `HierTS` under Eq. (26) by `HierTS-2`. Then, we start by highlighting the differences between these two variants of `HierTS`. First, their regret bounds scale as

$$\texttt{HierTS-1} : \tilde{\mathcal{O}}\Big(\sqrt{nd(K \sum_{\ell=1}^{L} \sigma_\ell^2 + L\sigma_{L+1}^2)}\Big), \quad \texttt{HierTS-2} : \tilde{\mathcal{O}}\Big(\sqrt{nd(K\sigma_1^2 + \sum_{\ell=1}^{L} \sigma_{\ell+1}^2)}\Big).$$

When $K \approx L$, the regret bounds of `HierTS-1` and `HierTS-2` are similar. However, when $K > L$, `HierTS-2` outperforms `HierTS-1`. This is because `HierTS-2` puts more uncertainty on a single $d$-dimensional latent parameter $\psi_1$, rather than $K$ individual $d$-dimensional action parameters $\theta_a$. More importantly, `HierTS-1` implicitly assumes that action parameters $\theta_a$ are conditionally independent given $\psi_L$, which is not true. Consequently, `HierTS-2` outperforms `HierTS-1`. Note that, under the linear diffusion model Eq. (17), dTS and `HierTS-2` have roughly similar regret bounds. Specifically, their regret bounds dependency on $K$ is identical, where both methods involve multiplying $K$ by $\sigma_1^2$, and both enjoy improved performance compared to `HierTS-1`. That said, note that Theorem E.1 and Proposition E.2 provide an understanding of how dTS's regret scales under linear link functions $f_\ell$, and do not say that using dTS is better than using `HierTS` when the link functions $f_\ell$ are linear since the latter can be obtained by a proper marginalization of latent parameters (i.e., `HierTS-2` instead of `HierTS-1`). While such a comparison is not the goal of this work, we still provide it for completeness next.

When the mixing matrices $\mathrm{W}_\ell$ are dense (i.e., assumption **(A3)** is not applicable), dTS and `HierTS-2` have comparable regret bounds and computational efficiency. However, under the sparsity assumption **(A3)** and with mixing matrices that allow for conditional independence of $\psi_1$ coordinates given $\psi_2$, dTS enjoys a computational advantage over `HierTS-2`. This advantage explains why works focusing on multi-level hierarchies typically benchmark their algorithms against two-level structures akin to `HierTS-1`, rather than the more competitive `HierTS-2`. This is also consistent with prior works in Bayesian bandits using multi-level hierarchies, such as Tree-based priors [32], which compared their method to `HierTS-1`. In line with this, we also compared dTS with `HierTS-1` in our experiments. But this is only given for completeness as this is not the aim of Theorem E.1 and Proposition E.2. More importantly, `HierTS` is inapplicable in the general case in Eq. (1) with non-linear link functions since the latent parameters cannot be analytically marginalized.

## D.2   Additional discussion: why regret bound depends on $K$ and $L$

**Why the bound increases with $K$?** This arises due to our conditional learning of $\theta_a$ given $\psi_1$. Rather than assuming deterministic linearity, $\theta_a = \mathrm{W}_1 \psi_1$, we account for stochasticity by modeling $\theta_a \sim \mathcal{N}(\mathrm{W}_1 \psi_1, \sigma_1^2 I_d)$. This makes dTS robust to misspecification scenarios where $\theta_a$ is not perfectly linear with respect to $\psi_1$, at the cost of additional learning of $\theta_a \mid \psi_1$. If we were to assume deterministic linearity ($\sigma_1 = 0$), our regret bound would scale with $L$ only.

**Why the bound increases with $L$?** This is because increasing the number of layers $L$ adds more initial uncertainty due to the additional covariance introduced by the extra layers. However, this does not imply that we should always use $L = 1$ (the minimum possible $L$). Precisely, the theoretical results predict that regret increases with $L$ when the true prior distribution matches a diffusion model of depth $L$, as increasing $L$ reflects a more complex action parameter distribution and hence a more

complex bandit problem. However, in practice, when $L$ is small, the pre-trained diffusion model may be too simple to capture the true prior distribution, violating the assumptions of our theory. Increasing $L$ improves the pre-trained model's quality, reducing regret. Once $L$ is large enough and the pre-trained model adequately captures the true prior distribution, the theoretical assumptions hold, and regret begins to increase with $L$, as predicted. This is validated empirically in Fig. 3b.

# E   Formal theory

We analyze dTS assuming that: **(A1)** The rewards are linear $P(\cdot \mid x; \theta_a) = \mathcal{N}(\cdot; x^\top \theta_a, \sigma^2)$. **(A2)** The link functions $f_\ell$ are linear such as $f_\ell(\psi_\ell) = W_\ell \psi_\ell$ for $\ell \in [L]$, where $W_\ell \in \mathbb{R}^{d \times d}$ are *known mixing matrices*. This leads to a structure with $L$ layers of linear Gaussian relationships detailed in Appendix B.1. In particular, this leads to closed-form posteriors given in Appendix B.2 that inspired our approximation and enable theory similar to linear bandits [3]. However, proofs are not the same, and technical challenges remain (explained in Appendix F).

Although our result holds for milder assumptions, we make additional simplifications for clarity and interpretability. We assume that **(A3)** Contexts satisfy $\|X_t\|_2^2 = 1$ for any $t \in [n]$. Note that **(A3)** can be relaxed to any contexts $X_t$ with bounded norms $\|X_t\|_2$. **(A4)** Mixing matrices and covariances satisfy $\lambda_1(W_\ell^\top W_\ell) = 1$ for any $\ell \in [L]$ and $\Sigma_\ell = \sigma_\ell^2 I_d$ for any $\ell \in [L+1]$. **(A4)** can be relaxed to positive definite covariances $\Sigma_\ell$ and arbitrary mixing matrices $W_\ell$. In particular, this is satisfied once we use a diffusion model parametrized with linear functions. In this section, we write $\tilde{\mathcal{O}}$ for the big-O notation up to polylogarithmic factors. We start by stating our bound for dTS.

**Theorem E.1.** *Let* $\sigma_{\text{MAX}}^2 = \max_{\ell \in [L+1]} 1 + \frac{\sigma_\ell^2}{\sigma^2}$. *There exists a constant* $c > 0$ *such that for any* $\delta \in (0, 1)$, *the Bayes regret of* dTS *under* **(A1)**, **(A2)**, **(A3)** *and* **(A4)** *is bounded as*

$$\mathcal{BR}(n) \leq \sqrt{2n\big(\mathcal{R}^{\text{ACT}}(n) + \sum_{\ell=1}^{L} \mathcal{R}_\ell^{\text{LAT}}\big) \log(1/\delta)\big)} + cn\delta,$$

$$\mathcal{R}^{\text{ACT}}(n) = c_0 dK \log\big(1 + \frac{n\sigma_1^2}{d}\big), \ c_0 = \frac{\sigma_1^2}{\log(1 + \sigma_1^2)},$$

$$\mathcal{R}_\ell^{\text{LAT}} = c_\ell d \log\big(1 + \frac{\sigma_{\ell+1}^2}{\sigma_\ell^2}\big), c_\ell = \frac{\sigma_{\ell+1}^2 \sigma_{\text{MAX}}^{2\ell}}{\log(1 + \sigma_{\ell+1}^2)}, \tag{27}$$

Eq. (27) holds for any $\delta \in (0, 1)$. In particular, the term $cn\delta$ is constant when $\delta = 1/n$. Then, the bound is $\tilde{\mathcal{O}}\big(\sqrt{n(dK\sigma_1^2 + d\sum_{\ell=1}^{L} \sigma_{\ell+1}^2 \sigma_{\text{MAX}}^{2\ell})}\big)$, and this dependence on the horizon $n$ aligns with prior Bayes regret bounds. The bound comprises $L + 1$ main terms, $\mathcal{R}^{\text{ACT}}(n)$ and $\mathcal{R}_\ell^{\text{LAT}}$ for $\ell \in [L]$. First, $\mathcal{R}^{\text{ACT}}(n)$ relates to action parameters learning, conforming to a standard form [52]. Similarly, $\mathcal{R}_\ell^{\text{LAT}}$ is associated with learning the $\ell$-th latent parameter.

To include more structure, we propose the *sparsity* assumption **(A5)** $W_\ell = (\bar{W}_\ell, 0_{d, d-d_\ell})$, where $\bar{W}_\ell \in \mathbb{R}^{d \times d_\ell}$ for any $\ell \in [L]$. Note that **(A5)** is not an assumption when $d_\ell = d$ for any $\ell \in [L]$. Notably, **(A5)** incorporates a plausible structural characteristic that a diffusion model could capture.

**Proposition E.2** (Sparsity). *Let* $\sigma_{\text{MAX}}^2 = \max_{\ell \in [L+1]} 1 + \frac{\sigma_\ell^2}{\sigma^2}$. *There exists a constant* $c > 0$ *such that for any* $\delta \in (0, 1)$, *the Bayes regret of* dTS *under* **(A1)**, **(A2)**, **(A3)**, **(A4)** *and* **(A5)** *is bounded as*

$$\mathcal{BR}(n) \leq \sqrt{2n\big(\mathcal{R}^{\text{ACT}}(n) + \sum_{\ell=1}^{L} \tilde{\mathcal{R}}_\ell^{\text{LAT}}\big) \log(1/\delta)\big)} + cn\delta,$$

$$\mathcal{R}^{\text{ACT}}(n) = c_0 dK \log\big(1 + \frac{n\sigma_1^2}{d}\big), c_0 = \frac{\sigma_1^2}{\log(1 + \sigma_1^2)},$$

$$\tilde{\mathcal{R}}_\ell^{\text{LAT}} = c_\ell d_\ell \log\big(1 + \frac{\sigma_{\ell+1}^2}{\sigma_\ell^2}\big), c_\ell = \frac{\sigma_{\ell+1}^2 \sigma_{\text{MAX}}^{2\ell}}{\log(1 + \sigma_{\ell+1}^2)}. \tag{28}$$

From [Proposition E.2](), our bounds scales as

$$\mathcal{BR}(n) = \tilde{\mathcal{O}}\Big(\sqrt{n(dK\sigma_1^2 + \sum_{\ell=1}^{L} d_\ell \sigma_{\ell+1}^2 \sigma_{\text{MAX}}^{2\ell})}\Big). \tag{29}$$

The Bayes regret bound has a clear interpretation: if the true environment parameters are drawn from the prior, then the expected regret of an algorithm stays below that bound. Consequently, a less informative prior (such as high variance) leads to a more challenging problem and thus a higher bound. Then, smaller values of $K$, $L$, $d$ or $d_\ell$ translate to fewer parameters to learn, leading to lower regret. The regret also decreases when the initial variances $\sigma_\ell^2$ decrease. These dependencies are common in Bayesian analysis, and empirical results match them.

The reader might question the dependence of our bound on both $L$ and $K$. Details can be found in [Appendix G.5](), but in summary, we model the relationship between $\theta_a$ and $\psi_1$ stochastically as $\mathcal{N}(W_1\psi_1, \sigma_1^2 I_d)$ to account for potential nonlinearity. This choice makes the model robust to model misspecification but introduces extra uncertainty and requires learning both the $\theta_a$ and the $\psi_\ell$. This results in a regret bound that depends on both $K$ and $L$. However, thanks to the use of informative priors, our bound has significantly smaller constants compared to both the Bayesian regret for `LinTS` and its frequentist counterpart, as demonstrated empirically in [Appendix G.5]() where it is much tighter than both and in [Section 4.1]() where we theoretically compare our Bayes regret bound to that of `LinTS`.

**Technical contributions.** `dTS` uses hierarchical sampling. Thus the marginal posterior distribution of $\theta_a \mid H_t$ is not explicitly defined. The first contribution is deriving $\theta_a \mid H_t$ using the total covariance decomposition combined with an induction proof, as our posteriors were derived recursively. Unlike standard analyses where the posterior distribution of $\theta_a \mid H_t$ is predetermined due to the absence of latent parameters, our method necessitates this recursive total covariance decomposition. Moreover, in standard proofs, we need to quantify the increase in posterior precision for the action taken $A_t$ in each round $t \in [n]$. However, in `dTS`, our analysis extends beyond this. We not only quantify the posterior information gain for the taken action but also for every latent parameter, since they are also learned. To elaborate, we use our recursive posteriors that connect the posterior covariance of each latent parameter $\psi_\ell$ with the covariance of the posterior action parameters $\theta_a$. This allows us to propagate the information gain associated with the action taken in round $A_t$ to all latent parameters $\psi_\ell$, for $\ell \in [L]$ by induction. Details are given in [Appendix F]().

# F Regret proof

## F.1 Sketch of the proof

We start with the following standard lemma upon which we build our analysis [8].

**Lemma F.1.** *Assume that $p(\theta_a \mid H_t) = \mathcal{N}(\theta_a; \breve{\mu}_{t,a}, \breve{\Sigma}_{t,a})$ for any $a \in [K]$, then for any $\delta \in (0,1)$,*

$$\mathcal{BR}(n) \leq \sqrt{2n\log(1/\delta)}\sqrt{\mathbb{E}\left[\sum_{t=1}^{n} \|X_t\|_{\breve{\Sigma}_{t,A_t}}^2\right]} + cn\delta, \qquad \text{where } c > 0 \text{ is a constant}. \tag{30}$$

Applying [Lemma F.1]() requires proving that the *marginal* action-posterior densities of $\theta_a \mid H_t$ in [Eq. (3)]() are Gaussian and computing their covariances, while we only know the *conditional* action-posteriors $p(\theta_a \mid \psi_1, H_t)$ and latent-posteriors $p(\psi_{\ell-1} \mid \psi_\ell, H_t)$. This is achieved by leveraging the preservation properties of the family of Gaussian distributions [38] and the total covariance decomposition [66] which leads to the next lemma.

**Lemma F.2.** *Let $t \in [n]$ and $a \in [K]$, then the marginal covariance matrix $\breve{\Sigma}_{t,a}$ reads*

$$\breve{\Sigma}_{t,a} = \hat{\Sigma}_{t,a} + \sum_{\ell \in [L]} P_{a,\ell} \bar{\Sigma}_{t,\ell} P_{a,\ell}^\top, \quad \text{where } P_{a,\ell} = \hat{\Sigma}_{t,a} \Sigma_1^{-1} W_1 \prod_{i=1}^{\ell-1} \bar{\Sigma}_{t,i} \Sigma_{i+1}^{-1} W_{i+1}. \tag{31}$$

The marginal covariance matrix $\breve{\Sigma}_{t,a}$ in [Eq. (31)]() decomposes into $L + 1$ terms. The first term corresponds to the posterior uncertainty of $\theta_a \mid \psi_1$. The remaining $L$ terms capture the posterior uncertainties of $\psi_L$ and $\psi_{\ell-1} \mid \psi_\ell$ for $\ell \in [L]/\{1\}$. These are then used to quantify the posterior information gain of latent parameters after one round as follows.

**Lemma F.3** (Posterior information gain). *Let $t \in [n]$ and $\ell \in [L]$, then*

$$\bar{\Sigma}_{t+1,\ell}^{-1} - \bar{\Sigma}_{t,\ell}^{-1} \succeq \sigma^{-2}\sigma_{\text{MAX}}^{-2\ell}\mathrm{P}_{A_t,\ell}^\top X_t X_t^\top \mathrm{P}_{A_t,\ell}, \qquad \textit{where } \sigma_{\text{MAX}}^2 = \max_{\ell \in [L+1]} 1 + \frac{\sigma_\ell^2}{\sigma^2}. \qquad (32)$$

Finally, Lemma F.2 is used to decompose $\|X_t\|_{\hat{\Sigma}_{t,A_t}}^2$ in Eq. (30) into $L + 1$ terms. Each term is bounded thanks to Lemma F.3. This results in the Bayes regret bound in Theorem E.1.

## F.2 Technical contributions

Our main technical contributions are the following.

**Lemma F.2.** In `dTS`, sampling is done hierarchically, meaning the marginal posterior distribution of $\theta_a|H_t$ is not explicitly defined. Instead, we use the conditional posterior distribution of $\theta_a|H_t, \psi_1$. The first contribution was deriving $\theta_a|H_t$ using the total covariance decomposition combined with an induction proof, as our posteriors in Appendix B.2 were derived recursively. Unlike in Bayes regret analysis for standard Thompson sampling, where the posterior distribution of $\theta_a|H_t$ is predetermined due to the absence of latent parameters, our method necessitates this recursive total covariance decomposition, marking a first difference from the standard Bayesian proofs of Thompson sampling. Note that `HierTS`, which is developed for multi-task linear bandits, also employs total covariance decomposition, but it does so under the assumption of a single latent parameter; on which action parameters are centered. Our extension significantly differs as it is tailored for contextual bandits with multiple, successive levels of latent parameters, moving away from `HierTS`'s assumption of a 1-level structure. Roughly speaking, `HierTS` when applied to contextual would consider a single-level hierarchy, where $\theta_a|\psi_1 \sim \mathcal{N}(\psi_1, \Sigma_1)$ with $L = 1$. In contrast, our model proposes a multi-level hierarchy, where the first level is $\theta_a|\psi_1 \sim \mathcal{N}(W_1\psi_1, \Sigma_1)$. This also introduces a new aspect to our approach - the use of a linear function $W_1\psi_1$, as opposed to `HierTS`'s assumption where action parameters are centered directly on the latent parameter. Thus, while `HierTS` also uses the total covariance decomposition, our generalize it to multi-level hierarchies under $L$ linear functions $W_\ell\psi_\ell$, instead of a single-level hierarchy under a single identity function $\psi_1$.

**Lemma F.3.** In Bayes regret proofs for standard Thompson sampling, we often quantify the posterior information gain. This is achieved by monitoring the increase in posterior precision for the action taken $A_t$ in each round $t \in [n]$. However, in `dTS`, our analysis extends beyond this. We not only quantify the posterior information gain for the taken action but also for every latent parameter, since they are also learned. This lemma addresses this aspect. To elaborate, we use the recursive formulas in Appendix B.2 that connect the posterior covariance of each latent parameter $\psi_\ell$ with the covariance of the posterior action parameters $\theta_a$. This allows us to propagate the information gain associated with the action taken in round $A_t$ to all latent parameters $\psi_\ell$, for $\ell \in [L]$ by induction. This is a novel contribution, as it is not a feature of Bayes regret analyses in standard Thompson sampling.

**Proposition E.2.** Building upon the insights of Theorem E.1, we introduce the sparsity assumption **(A3)**. Under this assumption, we demonstrate that the Bayes regret outlined in Theorem E.1 can be significantly refined. Specifically, the regret becomes contingent on dimensions $d_\ell \leq d$, as opposed to relying on the entire dimension $d$. The underlying principle of this sparsity assumption is straightforward: the Bayes regret is influenced by the quantity of parameters that require learning. With the sparsity assumption, this number is reduced to less than $d$ for each latent parameter. To substantiate this claim, we revisit the proof of Theorem E.1 and modify a crucial equality. This adjustment results in a more precise representation by partitioning the covariance matrix of each latent parameter $\psi_\ell$ into blocks. These blocks comprise a $d_\ell \times d_\ell$ segment corresponding to the learnable $d_\ell$ parameters of $\psi_\ell$, and another block of size $(d - d_\ell) \times (d - d_\ell)$ that does not necessitate learning. This decomposition allows us to conclude that the final regret is solely dependent on $d_\ell$, marking a significant refinement from the original theorem.

### F.3 Proof of lemma F.2

In this proof, we heavily rely on the total covariance decomposition [66]. Also, refer to [33, Section 5.2] for a brief introduction to this decomposition. Now, from Eq. (20), we have that

$$\text{cov}\left[\theta_a \mid H_t, \psi_1\right] = \hat{\Sigma}_{t,a} = \left(\hat{G}_{t,a} + \Sigma_1^{-1}\right)^{-1},$$

$$\mathbb{E}\left[\theta_a \mid H_t, \psi_1\right] = \hat{\mu}_{t,a} = \hat{\Sigma}_{t,a}\left(\hat{G}_{t,a}\hat{B}_{t,a} + \Sigma_1^{-1}W_1\psi_1\right).$$

First, given $H_t$, $\text{cov}\left[\theta_a \mid H_t, \psi_1\right] = \left(\hat{G}_{t,a} + \Sigma_1^{-1}\right)^{-1}$ is constant. Thus

$$\mathbb{E}\left[\text{cov}\left[\theta_a \mid H_t, \psi_1\right] \mid H_t\right] = \text{cov}\left[\theta_a \mid H_t, \psi_1\right] = \left(\hat{G}_{t,a} + \Sigma_1^{-1}\right)^{-1} = \hat{\Sigma}_{t,a}.$$

In addition, given $H_t$, $\hat{\Sigma}_{t,a}$, $\hat{G}_{t,a}$ and $\hat{B}_{t,a}$ are constant. Thus

$$\begin{aligned}
\text{cov}\left[\mathbb{E}\left[\theta_a \mid H_t, \psi_1\right] \mid H_t\right] &= \text{cov}\left[\hat{\Sigma}_{t,a}\left(\hat{G}_{t,a}\hat{B}_{t,a} + \Sigma_1^{-1}W_1\psi_1\right) \Big| H_t\right], \\
&= \text{cov}\left[\hat{\Sigma}_{t,a}\Sigma_1^{-1}W_1\psi_1 \Big| H_t\right], \\
&= \hat{\Sigma}_{t,a}\Sigma_1^{-1}W_1\text{cov}\left[\psi_1 \mid H_t\right]W_1^\top\Sigma_1^{-1}\hat{\Sigma}_{t,a}, \\
&= \hat{\Sigma}_{t,a}\Sigma_1^{-1}W_1\bar{\bar{\Sigma}}_{t,1}W_1^\top\Sigma_1^{-1}\hat{\Sigma}_{t,a},
\end{aligned}$$

where $\bar{\bar{\Sigma}}_{t,1} = \text{cov}\left[\psi_1 \mid H_t\right]$ is the marginal posterior covariance of $\psi_1$. Finally, the total covariance decomposition [66, 33] yields that

$$\begin{aligned}
\check{\Sigma}_{t,a} = \text{cov}\left[\theta_a \mid H_t\right] &= \mathbb{E}\left[\text{cov}\left[\theta_a \mid H_t, \psi_1\right] \mid H_t\right] + \text{cov}\left[\mathbb{E}\left[\theta_a \mid H_t, \psi_1\right] \mid H_t\right], \\
&= \hat{\Sigma}_{t,a} + \hat{\Sigma}_{t,a}\Sigma_1^{-1}W_1\bar{\bar{\Sigma}}_{t,1}W_1^\top\Sigma_1^{-1}\hat{\Sigma}_{t,a},
\end{aligned} \tag{33}$$

However, $\bar{\bar{\Sigma}}_{t,1} = \text{cov}\left[\psi_1 \mid H_t\right]$ is different from $\bar{\Sigma}_{t,1} = \text{cov}\left[\psi_1 \mid H_t, \psi_2\right]$ that we already derived in Eq. (21). Thus we do not know the expression of $\bar{\bar{\Sigma}}_{t,1}$. But we can use the same total covariance decomposition trick to find it. Precisely, let $\bar{\bar{\Sigma}}_{t,\ell} = \text{cov}\left[\psi_\ell \mid H_t\right]$ for any $\ell \in [L]$. Then we have that

$$\bar{\Sigma}_{t,1} = \text{cov}\left[\psi_1 \mid H_t, \psi_2\right] = \left(\Sigma_2^{-1} + \bar{G}_{t,1}\right)^{-1},$$

$$\bar{\mu}_{t,1} = \mathbb{E}\left[\psi_1 \mid H_t, \psi_2\right] = \bar{\Sigma}_{t,1}\left(\Sigma_2^{-1}W_2\psi_2 + \bar{B}_{t,1}\right).$$

First, given $H_t$, $\text{cov}\left[\psi_1 \mid H_t, \psi_2\right] = \left(\Sigma_2^{-1} + \bar{G}_{t,1}\right)^{-1}$ is constant. Thus

$$\mathbb{E}\left[\text{cov}\left[\psi_1 \mid H_t, \psi_2\right] \mid H_t\right] = \text{cov}\left[\psi_1 \mid H_t, \psi_2\right] = \bar{\Sigma}_{t,1}.$$

In addition, given $H_t$, $\bar{\Sigma}_{t,1}$, $\tilde{\Sigma}_{t,1}$ and $\bar{B}_{t,1}$ are constant. Thus

$$\begin{aligned}
\text{cov}\left[\mathbb{E}\left[\psi_1 \mid H_t, \psi_2\right] \mid H_t\right] &= \text{cov}\left[\bar{\Sigma}_{t,1}\left(\Sigma_2^{-1}W_2\psi_2 + \bar{B}_{t,1}\right) \Big| H_t\right], \\
&= \text{cov}\left[\bar{\Sigma}_{t,1}\Sigma_2^{-1}W_2\psi_2 \mid H_t\right], \\
&= \bar{\Sigma}_{t,1}\Sigma_2^{-1}W_2\text{cov}\left[\psi_2 \mid H_t\right]W_2^\top\Sigma_2^{-1}\bar{\Sigma}_{t,1}, \\
&= \bar{\Sigma}_{t,1}\Sigma_2^{-1}W_2\bar{\bar{\Sigma}}_{t,2}W_2^\top\Sigma_2^{-1}\bar{\Sigma}_{t,1}.
\end{aligned}$$

Finally, total covariance decomposition [66, 33] leads to

$$\begin{aligned}
\bar{\bar{\Sigma}}_{t,1} = \text{cov}\left[\psi_1 \mid H_t\right] &= \mathbb{E}\left[\text{cov}\left[\psi_1 \mid H_t, \psi_2\right] \mid H_t\right] + \text{cov}\left[\mathbb{E}\left[\psi_1 \mid H_t, \psi_2\right] \mid H_t\right], \\
&= \bar{\Sigma}_{t,1} + \bar{\Sigma}_{t,1}\Sigma_2^{-1}W_2\bar{\bar{\Sigma}}_{t,2}W_2^\top\Sigma_2^{-1}\bar{\Sigma}_{t,1}.
\end{aligned}$$

Now using the techniques, this can be generalized using the same technique as above to

$$\bar{\bar{\Sigma}}_{t,\ell} = \bar{\Sigma}_{t,\ell} + \bar{\Sigma}_{t,\ell}\Sigma_{\ell+1}^{-1}W_{\ell+1}\bar{\bar{\Sigma}}_{t,\ell+1}W_{\ell+1}^\top\Sigma_{\ell+1}^{-1}\bar{\Sigma}_{t,\ell}, \qquad \forall \ell \in [L-1].$$

Then, by induction, we get that

$$\bar{\bar{\Sigma}}_{t,1} = \sum_{\ell \in [L]} \bar{P}_\ell \bar{\Sigma}_{t,\ell} \bar{P}_\ell^\top, \qquad\qquad \forall \ell \in [L-1],$$

where we use that by definition $\bar{\bar{\Sigma}}_{t,L} = \mathrm{cov}\,[\psi_L \,|\, H_t] = \bar{\Sigma}_{t,L}$ and set $\bar{P}_1 = I_d$ and $\bar{P}_\ell = \prod_{i=1}^{\ell-1} \bar{\Sigma}_{t,i} \Sigma_{i+1}^{-1} W_{i+1}$ for any $\ell \in [L]/\{1\}$. Plugging this in Eq. (33) leads to

$$\begin{aligned}
\check{\Sigma}_{t,a} &= \hat{\Sigma}_{t,a} + \sum_{\ell \in [L]} \hat{\Sigma}_{t,a} \Sigma_1^{-1} W_1 \bar{P}_\ell \bar{\Sigma}_{t,\ell} \bar{P}_\ell^\top W_1^\top \Sigma_1^{-1} \hat{\Sigma}_{t,a}, \\
&= \hat{\Sigma}_{t,a} + \sum_{\ell \in [L]} \hat{\Sigma}_{t,a} \Sigma_1^{-1} W_1 \bar{P}_\ell \bar{\Sigma}_{t,\ell} (\hat{\Sigma}_{t,a} \Sigma_1^{-1} W_1)^\top, \\
&= \hat{\Sigma}_{t,a} + \sum_{\ell \in [L]} P_{a,\ell} \bar{\Sigma}_{t,\ell} P_{a,\ell}^\top,
\end{aligned}$$

where $P_{a,\ell} = \hat{\Sigma}_{t,a} \Sigma_1^{-1} W_1 \bar{P}_\ell = \hat{\Sigma}_{t,a} \Sigma_1^{-1} W_1 \prod_{i=1}^{\ell-1} \bar{\Sigma}_{t,i} \Sigma_{i+1}^{-1} W_{i+1}$.

## F.4 Proof of lemma F.3

We prove this result by induction. We start with the base case when $\ell = 1$.

**(I) Base case.** Let $u = \sigma^{-1} \hat{\Sigma}_{t,A_t}^{\frac{1}{2}} X_t$ From the expression of $\bar{\Sigma}_{t,1}$ in Eq. (21), we have that

$$\begin{aligned}
\bar{\Sigma}_{t+1,1}^{-1} - \bar{\Sigma}_{t,1}^{-1} &= W_1^\top \left( \Sigma_1^{-1} - \Sigma_1^{-1} (\hat{\Sigma}_{t,A_t}^{-1} + \sigma^{-2} X_t X_t^\top)^{-1} \Sigma_1^{-1} - (\Sigma_1^{-1} - \Sigma_1^{-1} \hat{\Sigma}_{t,A_t} \Sigma_1^{-1}) \right) W_1, \\
&= W_1^\top \left( \Sigma_1^{-1} (\hat{\Sigma}_{t,A_t} - (\hat{\Sigma}_{t,A_t}^{-1} + \sigma^{-2} X_t X_t^\top)^{-1}) \Sigma_1^{-1} \right) W_1, \\
&= W_1^\top \left( \Sigma_1^{-1} \hat{\Sigma}_{t,A_t}^{\frac{1}{2}} (I_d - (I_d + \sigma^{-2} \hat{\Sigma}_{t,A_t}^{\frac{1}{2}} X_t X_t^\top \hat{\Sigma}_{t,A_t}^{\frac{1}{2}})^{-1}) \hat{\Sigma}_{t,A_t}^{\frac{1}{2}} \Sigma_1^{-1} \right) W_1, \\
&= W_1^\top \left( \Sigma_1^{-1} \hat{\Sigma}_{t,A_t}^{\frac{1}{2}} (I_d - (I_d + uu^\top)^{-1}) \hat{\Sigma}_{t,A_t}^{\frac{1}{2}} \Sigma_1^{-1} \right) W_1, \\
&\overset{(i)}{=} W_1^\top \left( \Sigma_1^{-1} \hat{\Sigma}_{t,A_t}^{\frac{1}{2}} \frac{uu^\top}{1 + u^\top u} \hat{\Sigma}_{t,A_t}^{\frac{1}{2}} \Sigma_1^{-1} \right) W_1, \\
&\overset{(ii)}{=} \sigma^{-2} W_1^\top \Sigma_1^{-1} \hat{\Sigma}_{t,A_t} \frac{X_t X_t^\top}{1 + u^\top u} \hat{\Sigma}_{t,A_t} \Sigma_1^{-1} W_1.
\end{aligned} \tag{34}$$

In $(i)$ we use the Sherman-Morrison formula. Note that $(ii)$ says that $\bar{\Sigma}_{t+1,1}^{-1} - \bar{\Sigma}_{t,1}^{-1}$ is one-rank which we will also need in induction step. Now, we have that $\|X_t\|^2 = 1$. Therefore,

$$1 + u^\top u = 1 + \sigma^{-2} X_t^\top \hat{\Sigma}_{t,A_t} X_t \le 1 + \sigma^{-2} \lambda_1(\Sigma_1) \|X_t\|^2 = 1 + \sigma^{-2} \sigma_1^2 \le \sigma_{\mathrm{MAX}}^2,$$

where we use that by definition of $\sigma_{\mathrm{MAX}}^2$ in Lemma F.3, we have that $\sigma_{\mathrm{MAX}}^2 \ge 1 + \sigma^{-2} \sigma_1^2$. Therefore, by taking the inverse, we get that $\frac{1}{1+u^\top u} \ge \sigma_{\mathrm{MAX}}^{-2}$. Combining this with Eq. (34) leads to

$$\bar{\Sigma}_{t+1,1}^{-1} - \bar{\Sigma}_{t,1}^{-1} \succeq \sigma^{-2} \sigma_{\mathrm{MAX}}^{-2} W_1^\top \Sigma_1^{-1} \hat{\Sigma}_{t,A_t} X_t X_t^\top \hat{\Sigma}_{t,A_t} \Sigma_1^{-1} W_1$$

Noticing that $P_{A_t,1} = \hat{\Sigma}_{t,A_t} \Sigma_1^{-1} W_1$ concludes the proof of the base case when $\ell = 1$.

**(II) Induction step.** Let $\ell \in [L]/\{1\}$ and suppose that $\bar{\Sigma}_{t+1,\ell-1}^{-1} - \bar{\Sigma}_{t,\ell-1}^{-1}$ is one-rank and that it holds for $\ell - 1$ that

$$\bar{\Sigma}_{t+1,\ell-1}^{-1} - \bar{\Sigma}_{t,\ell-1}^{-1} \succeq \sigma^{-2} \sigma_{\mathrm{MAX}}^{-2(\ell-1)} P_{A_t,\ell-1}^\top X_t X_t^\top P_{A_t,\ell-1}, \quad \text{where } \sigma_{\mathrm{MAX}}^{-2} = \max_{\ell \in [L]} 1 + \sigma^{-2} \sigma_\ell^2.$$

Then, we want to show that $\bar{\Sigma}_{t+1,\ell}^{-1} - \bar{\Sigma}_{t,\ell}^{-1}$ is also one-rank and that it holds that

$$\bar{\Sigma}_{t+1,\ell}^{-1} - \bar{\Sigma}_{t,\ell}^{-1} \succeq \sigma^{-2} \sigma_{\mathrm{MAX}}^{-2\ell} P_{A_t,\ell}^\top X_t X_t^\top P_{A_t,\ell}, \qquad \text{where } \sigma_{\mathrm{MAX}}^{-2} = \max_{\ell \in [L]} 1 + \sigma^{-2} \sigma_\ell^2.$$

This is achieved as follows. First, we notice that by the induction hypothesis, we have that $\tilde{\Sigma}^{-1}_{t+1,\ell-1} - \bar{G}_{t,\ell-1} = \bar{\Sigma}^{-1}_{t+1,\ell-1} - \bar{\Sigma}^{-1}_{t,\ell-1}$ is one-rank. In addition, the matrix is positive semi-definite. Thus we can write it as $\tilde{\Sigma}^{-1}_{t+1,\ell-1} - \bar{G}_{t,\ell-1} = uu^\top$ where $u \in \mathbb{R}^d$. Then, similarly to the base case, we have

$$
\begin{aligned}
\bar{\Sigma}^{-1}_{t+1,\ell} - \bar{\Sigma}^{-1}_{t,\ell} &= \tilde{\Sigma}^{-1}_{t+1,\ell} - \tilde{\Sigma}^{-1}_{t,\ell}, \\
&= W_\ell^\top \left(\Sigma_\ell + \tilde{\Sigma}_{t+1,\ell-1}\right)^{-1} W_\ell - W_\ell^\top \left(\Sigma_\ell + \tilde{\Sigma}_{t,\ell-1}\right)^{-1} W_\ell, \\
&= W_\ell^\top \left[\left(\Sigma_\ell + \tilde{\Sigma}_{t+1,\ell-1}\right)^{-1} - \left(\Sigma_\ell + \tilde{\Sigma}_{t,\ell-1}\right)^{-1}\right] W_\ell, \\
&= W_\ell^\top \Sigma_\ell^{-1} \left[\left(\Sigma_\ell^{-1} + \bar{G}_{t,\ell-1}\right)^{-1} - \left(\Sigma_\ell^{-1} + \tilde{\Sigma}^{-1}_{t+1,\ell-1}\right)^{-1}\right] \Sigma_\ell^{-1} W_\ell, \\
&= W_\ell^\top \Sigma_\ell^{-1} \left[\left(\Sigma_\ell^{-1} + \bar{G}_{t,\ell-1}\right)^{-1} - \left(\Sigma_\ell^{-1} + \bar{G}_{t,\ell-1} + \tilde{\Sigma}^{-1}_{t+1,\ell-1} - \bar{G}_{t,\ell-1}\right)^{-1}\right] \Sigma_\ell^{-1} W_\ell, \\
&= W_\ell^\top \Sigma_\ell^{-1} \left[\left(\Sigma_\ell^{-1} + \bar{G}_{t,\ell-1}\right)^{-1} - \left(\Sigma_\ell^{-1} + \bar{G}_{t,\ell-1} + uu^\top\right)^{-1}\right] \Sigma_\ell^{-1} W_\ell, \\
&= W_\ell^\top \Sigma_\ell^{-1} \left[\bar{\Sigma}_{t,\ell-1} - \left(\bar{\Sigma}^{-1}_{t,\ell-1} + uu^\top\right)^{-1}\right] \Sigma_\ell^{-1} W_\ell, \\
&= W_\ell^\top \Sigma_\ell^{-1} \left[\bar{\Sigma}_{t,\ell-1} \frac{uu^\top}{1 + u^\top \bar{\Sigma}_{t,\ell-1} u} \bar{\Sigma}_{t,\ell-1}\right] \Sigma_\ell^{-1} W_\ell, \\
&= W_\ell^\top \Sigma_\ell^{-1} \bar{\Sigma}_{t,\ell-1} \frac{uu^\top}{1 + u^\top \bar{\Sigma}_{t,\ell-1} u} \bar{\Sigma}_{t,\ell-1} \Sigma_\ell^{-1} W_\ell
\end{aligned}
$$

However, we it follows from the induction hypothesis that $uu^\top = \tilde{\Sigma}^{-1}_{t+1,\ell-1} - \bar{G}_{t,\ell-1} = \bar{\Sigma}^{-1}_{t+1,\ell-1} - \bar{\Sigma}^{-1}_{t,\ell-1} \succeq \sigma^{-2} \sigma_{\text{MAX}}^{-2(\ell-1)} P^\top_{A_t,\ell-1} X_t X_t^\top P_{A_t,\ell-1}$. Therefore,

$$
\begin{aligned}
\bar{\Sigma}^{-1}_{t+1,\ell} - \bar{\Sigma}^{-1}_{t,\ell} &= W_\ell^\top \Sigma_\ell^{-1} \bar{\Sigma}_{t,\ell-1} \frac{uu^\top}{1 + u^\top \bar{\Sigma}_{t,\ell-1} u} \bar{\Sigma}_{t,\ell-1} \Sigma_\ell^{-1} W_\ell, \\
&\succeq W_\ell^\top \Sigma_\ell^{-1} \bar{\Sigma}_{t,\ell-1} \frac{\sigma^{-2} \sigma_{\text{MAX}}^{-2(\ell-1)} P^\top_{A_t,\ell-1} X_t X_t^\top P_{A_t,\ell-1}}{1 + u^\top \bar{\Sigma}_{t,\ell-1} u} \bar{\Sigma}_{t,\ell-1} \Sigma_\ell^{-1} W_\ell, \\
&= \frac{\sigma^{-2} \sigma_{\text{MAX}}^{-2(\ell-1)}}{1 + u^\top \bar{\Sigma}_{t,\ell-1} u} W_\ell^\top \Sigma_\ell^{-1} \bar{\Sigma}_{t,\ell-1} P^\top_{A_t,\ell-1} X_t X_t^\top P_{A_t,\ell-1} \bar{\Sigma}_{t,\ell-1} \Sigma_\ell^{-1} W_\ell, \\
&= \frac{\sigma^{-2} \sigma_{\text{MAX}}^{-2(\ell-1)}}{1 + u^\top \bar{\Sigma}_{t,\ell-1} u} P^\top_{A_t,\ell} X_t X_t^\top P_{A_t,\ell}.
\end{aligned}
$$

Finally, we use that $1 + u^\top \bar{\Sigma}_{t,\ell-1} u \leq 1 + \|u\|_2 \lambda_1(\bar{\Sigma}_{t,\ell-1}) \leq 1 + \sigma^{-2} \sigma_\ell^2$. Here we use that $\|u\|_2 \leq \sigma^{-2}$, which can also be proven by induction, and that $\lambda_1(\bar{\Sigma}_{t,\ell-1}) \leq \sigma_\ell^2$, which follows from the expression of $\bar{\Sigma}_{t,\ell-1}$ in [Appendix B.2](#). Therefore, we have that

$$
\begin{aligned}
\bar{\Sigma}^{-1}_{t+1,\ell} - \bar{\Sigma}^{-1}_{t,\ell} &\succeq \frac{\sigma^{-2} \sigma_{\text{MAX}}^{-2(\ell-1)}}{1 + u^\top \bar{\Sigma}_{t,\ell-1} u} P^\top_{A_t,\ell} X_t X_t^\top P_{A_t,\ell}, \\
&\succeq \frac{\sigma^{-2} \sigma_{\text{MAX}}^{-2(\ell-1)}}{1 + \sigma^{-2} \sigma_\ell^2} P^\top_{A_t,\ell} X_t X_t^\top P_{A_t,\ell}, \\
&\succeq \sigma^{-2} \sigma_{\text{MAX}}^{-2\ell} P^\top_{A_t,\ell} X_t X_t^\top P_{A_t,\ell},
\end{aligned}
$$

where the last inequality follows from the definition of $\sigma_{\text{MAX}}^2 = \max_{\ell \in [L]} 1 + \sigma^{-2} \sigma_\ell^2$. This concludes the proof.

### F.5 Proof of theorem [E.1](#)

We start with the following standard result which we borrow from [32, 8],

$$
\mathcal{BR}(n) \leq \sqrt{2n \log(1/\delta)} \sqrt{\mathbb{E}\left[\sum_{t=1}^n \|X_t\|^2_{\tilde{\Sigma}_{t,A_t}}\right]} + cn\delta, \qquad \text{where } c > 0 \text{ is a constant}. \tag{35}
$$

Then we use Lemma F.2 and express the marginal covariance $\check{\Sigma}_{t,A_t}$ as

$$\check{\Sigma}_{t,a} = \hat{\Sigma}_{t,a} + \sum_{\ell \in [L]} \mathrm{P}_{a,\ell} \bar{\Sigma}_{t,\ell} \mathrm{P}_{a,\ell}^\top, \qquad \text{where } \mathrm{P}_{a,\ell} = \hat{\Sigma}_{t,a} \Sigma_1^{-1} \mathrm{W}_1 \prod_{i=1}^{\ell-1} \bar{\Sigma}_{t,i} \Sigma_{i+1}^{-1} \mathrm{W}_{i+1}. \qquad (36)$$

Therefore, we can decompose $\|X_t\|_{\check{\Sigma}_{t,A_t}}^2$ as

$$\|X_t\|_{\check{\Sigma}_{t,A_t}}^2 = \sigma^2 \frac{X_t^\top \check{\Sigma}_{t,A_t} X_t}{\sigma^2} \overset{(i)}{=} \sigma^2 \Big( \sigma^{-2} X_t^\top \hat{\Sigma}_{t,A_t} X_t + \sigma^{-2} \sum_{\ell \in [L]} X_t^\top \mathrm{P}_{A_t,\ell} \bar{\Sigma}_{t,\ell} \mathrm{P}_{A_t,\ell}^\top X_t \Big),$$

$$\overset{(ii)}{\leq} c_0 \log(1 + \sigma^{-2} X_t^\top \hat{\Sigma}_{t,A_t} X_t) + \sum_{\ell \in [L]} c_\ell \log(1 + \sigma^{-2} X_t^\top \mathrm{P}_{A_t,\ell} \bar{\Sigma}_{t,\ell} \mathrm{P}_{A_t,\ell}^\top X_t), \qquad (37)$$

where $(i)$ follows from Eq. (36), and we use the following inequality in $(ii)$

$$x = \frac{x}{\log(1+x)} \log(1+x) \leq \Big( \max_{x \in [0,u]} \frac{x}{\log(1+x)} \Big) \log(1+x) = \frac{u}{\log(1+u)} \log(1+x),$$

which holds for any $x \in [0, u]$, where constants $c_0$ and $c_\ell$ are derived as

$$c_0 = \frac{\sigma_1^2}{\log(1 + \frac{\sigma_1^2}{\sigma^2})}, \quad c_\ell = \frac{\sigma_{\ell+1}^2}{\log(1 + \frac{\sigma_{\ell+1}^2}{\sigma^2})}, \text{ with the convention that } \sigma_{L+1} = 1.$$

The derivation of $c_0$ uses that

$$X_t^\top \hat{\Sigma}_{t,A_t} X_t \leq \lambda_1(\hat{\Sigma}_{t,A_t}) \|X_t\|^2 \leq \lambda_d^{-1}(\Sigma_1^{-1} + G_{t,A_t}) \leq \lambda_d^{-1}(\Sigma_1^{-1}) = \lambda_1(\Sigma_1) = \sigma_1^2.$$

The derivation of $c_\ell$ follows from

$$X_t^\top \mathrm{P}_{A_t,\ell} \bar{\Sigma}_{t,\ell} \mathrm{P}_{A_t,\ell}^\top X_t \leq \lambda_1(\mathrm{P}_{A_t,\ell} \mathrm{P}_{A_t,\ell}^\top) \lambda_1(\bar{\Sigma}_{t,\ell}) \|X_t\|^2 \leq \sigma_{\ell+1}^2.$$

Therefore, from Eq. (37) and Eq. (35), we get that

$$\mathcal{BR}(n) \leq \sqrt{2n \log(1/\delta)} \Big( \mathbb{E} \Big[ c_0 \sum_{t=1}^n \log(1 + \sigma^{-2} X_t^\top \hat{\Sigma}_{t,A_t} X_t)$$

$$+ \sum_{\ell \in [L]} c_\ell \sum_{t=1}^n \log(1 + \sigma^{-2} X_t^\top \mathrm{P}_{A_t,\ell} \bar{\Sigma}_{t,\ell} \mathrm{P}_{A_t,\ell}^\top X_t) \Big] \Big)^{\frac{1}{2}} + cn\delta \qquad (38)$$

Now we focus on bounding the logarithmic terms in Eq. (38).

**(I) First term in Eq. (38)** We first rewrite this term as

$$\log(1 + \sigma^{-2} X_t^\top \hat{\Sigma}_{t,A_t} X_t) \overset{(i)}{=} \log \det(I_d + \sigma^{-2} \hat{\Sigma}_{t,A_t}^{\frac{1}{2}} X_t X_t^\top \hat{\Sigma}_{t,A_t}^{\frac{1}{2}}),$$

$$= \log \det(\hat{\Sigma}_{t,A_t}^{-1} + \sigma^{-2} X_t X_t^\top) - \log \det(\hat{\Sigma}_{t,A_t}^{-1}) = \log \det(\hat{\Sigma}_{t+1,A_t}^{-1}) - \log \det(\hat{\Sigma}_{t,A_t}^{-1}),$$

where $(i)$ follows from the Weinstein-Aronszajn identity. Then we sum over all rounds $t \in [n]$, and get a telescoping

$$\sum_{t=1}^n \log \det(I_d + \sigma^{-2} \hat{\Sigma}_{t,A_t}^{\frac{1}{2}} X_t X_t^\top \hat{\Sigma}_{t,A_t}^{\frac{1}{2}}) = \sum_{t=1}^n \log \det(\hat{\Sigma}_{t+1,A_t}^{-1}) - \log \det(\hat{\Sigma}_{t,A_t}^{-1}),$$

$$= \sum_{t=1}^n \sum_{a=1}^K \log \det(\hat{\Sigma}_{t+1,a}^{-1}) - \log \det(\hat{\Sigma}_{t,a}^{-1}) = \sum_{a=1}^K \sum_{t=1}^n \log \det(\hat{\Sigma}_{t+1,a}^{-1}) - \log \det(\hat{\Sigma}_{t,a}^{-1}),$$

$$= \sum_{a=1}^K \log \det(\hat{\Sigma}_{n+1,a}^{-1}) - \log \det(\hat{\Sigma}_{1,a}^{-1}) \overset{(i)}{=} \sum_{a=1}^K \log \det(\Sigma_1^{\frac{1}{2}} \hat{\Sigma}_{n+1,a}^{-1} \Sigma_1^{\frac{1}{2}}),$$

where $(i)$ follows from the fact that $\hat{\Sigma}_{1,a} = \Sigma_1$. Now we use the inequality of arithmetic and geometric means and get

$$\sum_{t=1}^{n} \log \det(I_d + \sigma^{-2}\hat{\Sigma}_{t,A_t}^{\frac{1}{2}} X_t X_t^{\top}\hat{\Sigma}_{t,A_t}^{\frac{1}{2}}) = \sum_{a=1}^{K} \log \det(\Sigma_1^{\frac{1}{2}}\hat{\Sigma}_{n+1,a}^{-1}\Sigma_1^{\frac{1}{2}}),$$

$$\leq \sum_{a=1}^{K} d \log\left(\frac{1}{d}\operatorname{Tr}(\Sigma_1^{\frac{1}{2}}\hat{\Sigma}_{n+1,a}^{-1}\Sigma_1^{\frac{1}{2}})\right), \tag{39}$$

$$\leq \sum_{a=1}^{K} d \log\left(1 + \frac{n}{d}\frac{\sigma_1^2}{\sigma^2}\right) = Kd \log\left(1 + \frac{n}{d}\frac{\sigma_1^2}{\sigma^2}\right).$$

**(II) Remaining terms in Eq. (38)** Let $\ell \in [L]$. Then we have that

$$\log(1 + \sigma^{-2}X_t^{\top}\mathrm{P}_{A_t,\ell}\bar{\Sigma}_{t,\ell}\mathrm{P}_{A_t,\ell}^{\top}X_t) = \sigma_{\mathrm{MAX}}^{2\ell}\sigma_{\mathrm{MAX}}^{-2\ell}\log(1 + \sigma^{-2}X_t^{\top}\mathrm{P}_{A_t,\ell}\bar{\Sigma}_{t,\ell}\mathrm{P}_{A_t,\ell}^{\top}X_t),$$

$$\leq \sigma_{\mathrm{MAX}}^{2\ell}\log(1 + \sigma^{-2}\sigma_{\mathrm{MAX}}^{-2\ell}X_t^{\top}\mathrm{P}_{A_t,\ell}\bar{\Sigma}_{t,\ell}\mathrm{P}_{A_t,\ell}^{\top}X_t),$$

$$\overset{(i)}{=} \sigma_{\mathrm{MAX}}^{2\ell}\log\det(I_d + \sigma^{-2}\sigma_{\mathrm{MAX}}^{-2\ell}\bar{\Sigma}_{t,\ell}^{\frac{1}{2}}\mathrm{P}_{A_t,\ell}^{\top}X_t X_t^{\top}\mathrm{P}_{A_t,\ell}\bar{\Sigma}_{t,\ell}^{\frac{1}{2}}),$$

$$= \sigma_{\mathrm{MAX}}^{2\ell}\left(\log\det(\bar{\Sigma}_{t,\ell}^{-1} + \sigma^{-2}\sigma_{\mathrm{MAX}}^{-2\ell}\mathrm{P}_{A_t,\ell}^{\top}X_t X_t^{\top}\mathrm{P}_{A_t,\ell}) - \log\det(\bar{\Sigma}_{t,\ell}^{-1})\right),$$

where we use the Weinstein-Aronszajn identity in $(i)$. Now we know from Lemma F.3 that the following inequality holds $\sigma^{-2}\sigma_{\mathrm{MAX}}^{-2\ell}\mathrm{P}_{A_t,\ell}^{\top}X_t X_t^{\top}\mathrm{P}_{A_t,\ell} \preceq \bar{\Sigma}_{t+1,\ell}^{-1} - \bar{\Sigma}_{t,\ell}^{-1}$. As a result, we get that $\bar{\Sigma}_{t,\ell}^{-1} + \sigma^{-2}\sigma_{\mathrm{MAX}}^{-2\ell}\mathrm{P}_{A_t,\ell}^{\top}X_t X_t^{\top}\mathrm{P}_{A_t,\ell} \preceq \bar{\Sigma}_{t+1,\ell}^{-1}$. Thus,

$$\log(1 + \sigma^{-2}X_t^{\top}\mathrm{P}_{A_t,\ell}\bar{\Sigma}_{t,\ell}\mathrm{P}_{A_t,\ell}^{\top}X_t) \leq \sigma_{\mathrm{MAX}}^{2\ell}\left(\log\det(\bar{\Sigma}_{t+1,\ell}^{-1}) - \log\det(\bar{\Sigma}_{t,\ell}^{-1})\right),$$

Then we sum over all rounds $t \in [n]$, and get a telescoping

$$\sum_{t=1}^{n} \log(1 + \sigma^{-2}X_t^{\top}\mathrm{P}_{A_t,\ell}\bar{\Sigma}_{t,\ell}\mathrm{P}_{A_t,\ell}^{\top}X_t) \leq \sigma_{\mathrm{MAX}}^{2\ell}\sum_{t=1}^{n} \log\det(\bar{\Sigma}_{t+1,\ell}^{-1}) - \log\det(\bar{\Sigma}_{t,\ell}^{-1}),$$

$$= \sigma_{\mathrm{MAX}}^{2\ell}\left(\log\det(\bar{\Sigma}_{n+1,\ell}^{-1}) - \log\det(\bar{\Sigma}_{1,\ell}^{-1})\right),$$

$$\overset{(i)}{=} \sigma_{\mathrm{MAX}}^{2\ell}\left(\log\det(\bar{\Sigma}_{n+1,\ell}^{-1}) - \log\det(\Sigma_{\ell+1}^{-1})\right),$$

$$= \sigma_{\mathrm{MAX}}^{2\ell}\left(\log\det(\Sigma_{\ell+1}^{\frac{1}{2}}\bar{\Sigma}_{n+1,\ell}^{-1}\Sigma_{\ell+1}^{\frac{1}{2}})\right),$$

where we use that $\bar{\Sigma}_{1,\ell} = \Sigma_{\ell+1}$ in $(i)$. Finally, we use the inequality of arithmetic and geometric means and get that

$$\sum_{t=1}^{n} \log(1 + \sigma^{-2}X_t^{\top}\mathrm{P}_{A_t,\ell}\bar{\Sigma}_{t,\ell}\mathrm{P}_{A_t,\ell}^{\top}X_t) \leq \sigma_{\mathrm{MAX}}^{2\ell}\left(\log\det(\Sigma_{\ell+1}^{\frac{1}{2}}\bar{\Sigma}_{n+1,\ell}^{-1}\Sigma_{\ell+1}^{\frac{1}{2}})\right),$$

$$\leq d\sigma_{\mathrm{MAX}}^{2\ell}\log\left(\frac{1}{d}\operatorname{Tr}(\Sigma_{\ell+1}^{\frac{1}{2}}\bar{\Sigma}_{n+1,\ell}^{-1}\Sigma_{\ell+1}^{\frac{1}{2}})\right), \tag{40}$$

$$\leq d\sigma_{\mathrm{MAX}}^{2\ell}\log\left(1 + \frac{\sigma_{\ell+1}^2}{\sigma_\ell^2}\right),$$

The last inequality follows from the expression of $\bar{\Sigma}_{n+1,\ell}^{-1}$ in Eq. (21) that leads to

$$\Sigma_{\ell+1}^{\frac{1}{2}}\bar{\Sigma}_{n+1,\ell}^{-1}\Sigma_{\ell+1}^{\frac{1}{2}} = I_d + \Sigma_{\ell+1}^{\frac{1}{2}}\bar{G}_{t,\ell}\Sigma_{\ell+1}^{\frac{1}{2}},$$

$$= I_d + \Sigma_{\ell+1}^{\frac{1}{2}}\mathrm{W}_\ell^{\top}\left(\Sigma_\ell^{-1} - \Sigma_\ell^{-1}\bar{\Sigma}_{t,\ell-1}\Sigma_\ell^{-1}\right)\mathrm{W}_\ell\Sigma_{\ell+1}^{\frac{1}{2}}, \tag{41}$$

since $\bar{G}_{t,\ell} = W_\ell^\top \big( \Sigma_\ell^{-1} - \Sigma_\ell^{-1} \bar{\Sigma}_{t,\ell-1} \Sigma_\ell^{-1} \big) W_\ell$. This allows us to bound $\frac{1}{d} \mathrm{Tr}(\Sigma_{\ell+1}^{\frac{1}{2}} \bar{\Sigma}_{n+1,\ell}^{-1} \Sigma_{\ell+1}^{\frac{1}{2}})$ as

$$
\begin{aligned}
\frac{1}{d} \mathrm{Tr}(\Sigma_{\ell+1}^{\frac{1}{2}} \bar{\Sigma}_{n+1,\ell}^{-1} \Sigma_{\ell+1}^{\frac{1}{2}}) &= \frac{1}{d} \mathrm{Tr}(I_d + \Sigma_{\ell+1}^{\frac{1}{2}} W_\ell^\top \big( \Sigma_\ell^{-1} - \Sigma_\ell^{-1} \bar{\Sigma}_{t,\ell-1} \Sigma_\ell^{-1} \big) W_\ell \Sigma_{\ell+1}^{\frac{1}{2}}) \,, \\
&= \frac{1}{d} (d + \mathrm{Tr}(\Sigma_{\ell+1}^{\frac{1}{2}} W_\ell^\top \big( \Sigma_\ell^{-1} - \Sigma_\ell^{-1} \bar{\Sigma}_{t,\ell-1} \Sigma_\ell^{-1} \big) W_\ell \Sigma_{\ell+1}^{\frac{1}{2}}) \,, \\
&\leq 1 + \frac{1}{d} \sum_{i=1}^d \lambda_1 (\Sigma_{\ell+1}^{\frac{1}{2}} W_\ell^\top \big( \Sigma_\ell^{-1} - \Sigma_\ell^{-1} \bar{\Sigma}_{t,\ell-1} \Sigma_\ell^{-1} \big) W_\ell \Sigma_{\ell+1}^{\frac{1}{2}} \,, \\
&\leq 1 + \frac{1}{d} \sum_{i=1}^d \lambda_1 (\Sigma_{\ell+1}) \lambda_1 (W_\ell^\top W_\ell) \lambda_1 \big( \Sigma_\ell^{-1} - \Sigma_\ell^{-1} \bar{\Sigma}_{t,\ell-1} \Sigma_\ell^{-1} \big) \,, \\
&\leq 1 + \frac{1}{d} \sum_{i=1}^d \lambda_1 (\Sigma_{\ell+1}) \lambda_1 (W_\ell^\top W_\ell) \lambda_1 \big( \Sigma_\ell^{-1} \big) \,, \\
&\leq 1 + \frac{1}{d} \sum_{i=1}^d \frac{\sigma_{\ell+1}^2}{\sigma_\ell^2} = 1 + \frac{\sigma_{\ell+1}^2}{\sigma_\ell^2} \,,
\end{aligned} \tag{42}
$$

where we use the assumption that $\lambda_1(W_\ell^\top W_\ell) = 1$ **(A2)** and that $\lambda_1(\Sigma_{\ell+1}) = \sigma_{\ell+1}^2$ and $\lambda_1(\Sigma_\ell^{-1}) = 1/\sigma_\ell^2$. This is because $\Sigma_\ell = \sigma_\ell^2 I_d$ for any $\ell \in [L+1]$. Finally, plugging Eqs. (39) and (40) in Eq. (38) concludes the proof.

### F.6  Proof of proposition E.2

We use exactly the same proof in Appendix F.5, with one change to account for the sparsity assumption **(A3)**. The change corresponds to Eq. (40). First, recall that Eq. (40) writes

$$
\sum_{t=1}^n \log(1 + \sigma^{-2} X_t^\top \mathrm{P}_{A_t,\ell} \bar{\Sigma}_{t,\ell} \mathrm{P}_{A_t,\ell}^\top X_t) \leq \sigma_{\mathrm{MAX}}^{2\ell} \Big( \log \det(\Sigma_{\ell+1}^{\frac{1}{2}} \bar{\Sigma}_{n+1,\ell}^{-1} \Sigma_{\ell+1}^{\frac{1}{2}}) \Big) \,,
$$

where

$$
\begin{aligned}
\Sigma_{\ell+1}^{\frac{1}{2}} \bar{\Sigma}_{n+1,\ell}^{-1} \Sigma_{\ell+1}^{\frac{1}{2}} &= I_d + \Sigma_{\ell+1}^{\frac{1}{2}} W_\ell^\top \big( \Sigma_\ell^{-1} - \Sigma_\ell^{-1} \bar{\Sigma}_{t,\ell-1} \Sigma_\ell^{-1} \big) W_\ell \Sigma_{\ell+1}^{\frac{1}{2}} \,, \\
&= I_d + \sigma_{\ell+1}^2 W_\ell^\top \big( \Sigma_\ell^{-1} - \Sigma_\ell^{-1} \bar{\Sigma}_{t,\ell-1} \Sigma_\ell^{-1} \big) W_\ell \,,
\end{aligned} \tag{43}
$$

where the second equality follows from the assumption that $\Sigma_{\ell+1} = \sigma_{\ell+1}^2 I_d$. But notice that in our assumption, **(A3)**, we assume that $W_\ell = (\bar{W}_\ell, 0_{d,d-d_\ell})$, where $\bar{W}_\ell \in \mathbb{R}^{d \times d_\ell}$ for any $\ell \in [L]$. Therefore, we have that for any $d \times d$ matrix $B \in \mathbb{R}^{dd \times d}$, the following holds, $W_\ell^\top B W_\ell = \begin{pmatrix} \bar{W}_\ell^\top B \bar{W}_\ell & 0_{d_\ell,d-d_\ell} \\ 0_{d-d_\ell,d_\ell} & 0_{d-d_\ell,d-d_\ell} \end{pmatrix}$. In particular, we have that

$$
W_\ell^\top \big( \Sigma_\ell^{-1} - \Sigma_\ell^{-1} \bar{\Sigma}_{t,\ell-1} \Sigma_\ell^{-1} \big) W_\ell = \begin{pmatrix} \bar{W}_\ell^\top \big( \Sigma_\ell^{-1} - \Sigma_\ell^{-1} \bar{\Sigma}_{t,\ell-1} \Sigma_\ell^{-1} \big) \bar{W}_\ell & 0_{d_\ell,d-d_\ell} \\ 0_{d-d_\ell,d_\ell} & 0_{d-d_\ell,d-d_\ell} \end{pmatrix} . \tag{44}
$$

Therefore, plugging this in Eq. (43) yields that

$$
\Sigma_{\ell+1}^{\frac{1}{2}} \bar{\Sigma}_{n+1,\ell}^{-1} \Sigma_{\ell+1}^{\frac{1}{2}} = \begin{pmatrix} I_{d_\ell} + \sigma_{\ell+1}^2 \bar{W}_\ell^\top \big( \Sigma_\ell^{-1} - \Sigma_\ell^{-1} \bar{\Sigma}_{t,\ell-1} \Sigma_\ell^{-1} \big) \bar{W}_\ell & 0_{d_\ell,d-d_\ell} \\ 0_{d-d_\ell,d_\ell} & I_{d-d_\ell} \end{pmatrix} . \tag{45}
$$

As a result, $\det(\Sigma_{\ell+1}^{\frac{1}{2}} \bar{\Sigma}_{n+1,\ell}^{-1} \Sigma_{\ell+1}^{\frac{1}{2}}) = \det(I_{d_\ell} + \sigma_{\ell+1}^2 \bar{W}_\ell^\top \big( \Sigma_\ell^{-1} - \Sigma_\ell^{-1} \bar{\Sigma}_{t,\ell-1} \Sigma_\ell^{-1} \big) \bar{W}_\ell)$. This allows us to move the problem from a $d$-dimensional one to a $d_\ell$-dimensional one. Then we use the inequality

of arithmetic and geometric means and get that

$$\sum_{t=1}^{n} \log(1 + \sigma^{-2} X_t^\top P_{A_t,\ell} \bar{\Sigma}_{t,\ell} P_{A_t,\ell}^\top X_t) \leq \sigma_{\text{MAX}}^{2\ell} \left( \log \det(\Sigma_{\ell+1}^{\frac{1}{2}} \bar{\Sigma}_{n+1,\ell}^{-1} \Sigma_{\ell+1}^{\frac{1}{2}}) \right),$$

$$= \sigma_{\text{MAX}}^{2\ell} \log \det(I_{d_\ell} + \sigma_{\ell+1}^2 \bar{W}_\ell^\top \left( \Sigma_\ell^{-1} - \Sigma_\ell^{-1} \bar{\Sigma}_{t,\ell-1} \Sigma_\ell^{-1} \right) \bar{W}_\ell),$$

$$\leq d_\ell \sigma_{\text{MAX}}^{2\ell} \log \left( \frac{1}{d_\ell} \text{Tr}(I_{d_\ell} + \sigma_{\ell+1}^2 \bar{W}_\ell^\top \left( \Sigma_\ell^{-1} - \Sigma_\ell^{-1} \bar{\Sigma}_{t,\ell-1} \Sigma_\ell^{-1} \right) \bar{W}_\ell) \right),$$

$$\leq d_\ell \sigma_{\text{MAX}}^{2\ell} \log \left( 1 + \frac{\sigma_{\ell+1}^2}{\sigma_\ell^2} \right). \tag{46}$$

To get the last inequality, we use derivations similar to the ones we used in Eq. (42). Finally, the desired result in obtained by replacing Eq. (40) by Eq. (46) in the previous proof in Appendix F.5.

# G    Additional experiments

## G.1    Swiss roll data

Fig. 4 shows samples from the Swiss roll data and samples from generated by the pre-trained diffusion model for different pre-training sample sizes.

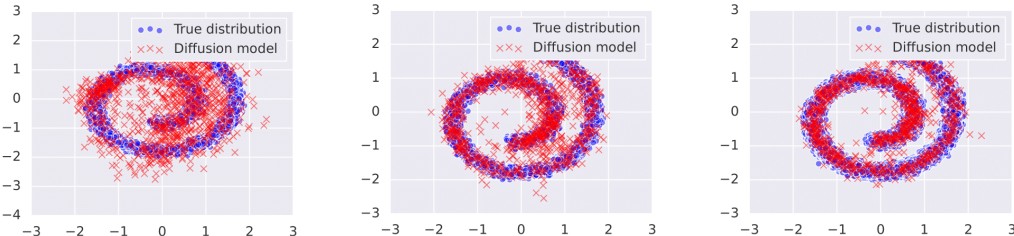

(a) Diffusion pre-trained on 50 samples from the Swiss roll dataset.

(b) Diffusion pre-trained on $10^3$ samples from the Swiss roll dataset.

(c) Diffusion pre-trained on $10^4$ samples from the Swiss roll dataset.

Figure 4: True distribution of action parameters (blue) vs. distribution of pre-trained diffusion model (red).

## G.2    Diffusion models pre-training

We used JAX for diffusion model pre-training, summarized as follows:

- **Parameterization:** Functions $f_\ell$ are parameterized with a fully connected 2-layer neural network (NN) with ReLU activation. The step $\ell$ is provided as input to capture the current sampling stage. Covariances are fixed (not learned) as $\Sigma_\ell = \sigma_\ell^2 I_d$ with $\sigma_\ell$ increasing with $\ell$.

- **Loss:** Offline data samples are progressively noised over steps $\ell \in [L]$, creating increasingly noisy versions of the data following a predefined noise schedule [30]. The NN is trained to reverse this noise (i.e., denoise) by predicting the noise added at each step. The loss function measures the $L_2$ norm difference between the predicted and actual noise at each step, as explained in Ho et al. [30].

- **Optimization:** Adam optimizer with a $10^{-3}$ learning rate was used. The NN was trained for 20,000 epochs with a batch size of min(2048, pre-training sample size). We used CPUs for pre-training, which was efficient enough to conduct multiple ablation studies.

- **After pre-training:** The pre-trained diffusion model is used as a prior for dTS and compared to LinTS as the reference baseline. In our ablation study, we plot the cumulative regret of LinTS in the last round divided by that of dTS. A ratio greater than 1 indicates that dTS outperforms LinTS, with higher values representing a larger performance gap.

### G.3 Quality of our posterior approximation

To assess the quality of our posterior approximation, we consider the scenario where the true distribution of action parameters is $\mathcal{N}(0_d, I_d)$ with $d = 2$ and rewards are linear. We pre-train a diffusion model using samples drawn from $\mathcal{N}(0_d, I_d)$. We then consider two priors: the true prior $\mathcal{N}(0_d, I_d)$ and the pre-trained diffusion model prior. This yields two posteriors:

- $P_1$ : Uses $\mathcal{N}(0_d, I_d)$ as the prior. $P_1$ is an exact posterior since the prior is Gaussian and rewards are linear-Gaussian.
- $P_2$ : Uses the pre-trained diffusion model as the prior. $P_2$ is our approximate posterior.

The learned diffusion model prior matches the true Gaussian prior (as seen in Fig. 5a). Thus, if our approximation is accurate, their posteriors $P_1$ and $P_2$ should also be similar. This is observed in Fig. 5b where the approximate posterior $P_2$ nearly matches the exact posterior $P_1$.

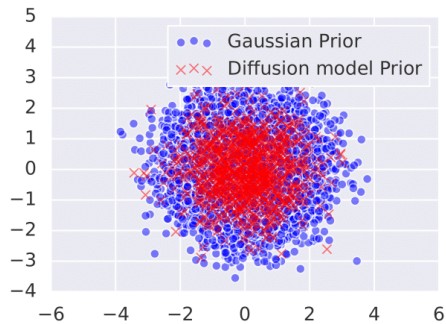 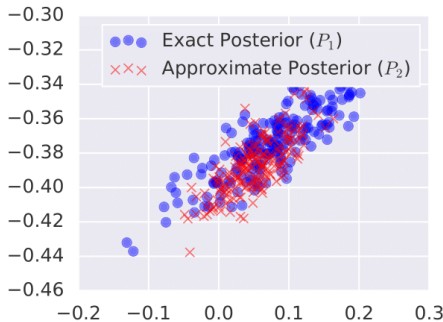

(a) Gaussian distribution vs. diffusion model pre-trained on $10^3$ samples drawn from it.

(b) Exact posterior $P_1$ vs. approximate posterior $P_2$ after $n = 100$ rounds of interactions.

Figure 5: Assessing the quality of our posterior approximation.

**Empirical posterior validation with MCMC.** Above, we assessed our posterior approximation in a tractable linear-Gaussian case, showing that the diffusion-based posterior closely matches the exact posterior. To further validate our approximation in more complex, non-linear settings, we compare our diffusion Thompson sampling (dTS) posterior samples to those obtained via two MCMC variants on the non-linear MovieLens benchmark:

- **MCMC-Fast:** Uses fewer sampling steps for efficiency.
- **MCMC-Slow:** Uses more sampling steps for higher accuracy.

As shown in Table 2, even the high-compute MCMC variant yields higher regret than dTS, motivating our efficient approximation for online bandits.

Table 2: Comparison between dTS and MCMC-based posteriors on MovieLens.

| Baseline | Regret Improvement (%) | Time Speed-Up (%) |
|---|---|---|
| dTS vs. MCMC-Fast | 50.6 % | 47.6 % |
| dTS vs. MCMC-Slow | 12.7 % | 80.5 % |

### G.4 CIFAR-10 ablation study

**CIFAR-10.** In Fig. 3a in Section 5.2, we showed that with only 10 pre-training samples, dTS outperforms LinTS on the Swiss-roll benchmark. We now extend this analysis to the vision dataset CIFAR-10 [40] (similar results were obtained on MNIST [48]). Our setting is similar to that in Hong et al. [32] and we use dTS's variant that uses a single shared parameter $\theta \in \mathbb{R}^d$ (Remark 2.1 and Section 3.3) because it is more suited for this setting. These additional ablations on CIFAR-10

confirm that `dTS` consistently benefits from offline pre-training, even when the true prior is not a diffusion model. Specifically, we vary the percentage of offline data used to train the prior and compare against both `HierTS` and `LinTS`.

Table 3: Regret improvement (%) of `dTS` on CIFAR-10.

| Offline Data (%) | vs. `HierTS` | vs. `LinTS` |
|---|---|---|
| 1% | 69.11% | 87.74% |
| 5% | 79.56% | 92.18% |
| 25% | 80.65% | 92.48% |
| 50% | 81.67% | 92.88% |

### G.5 Bound comparison

Here, we compare our bound in Theorem E.1 to bounds of `LinTS` from the literature.

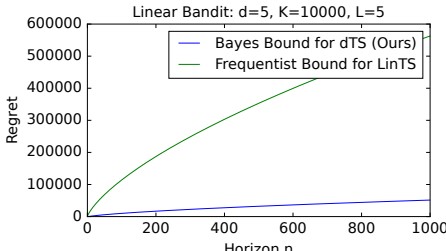
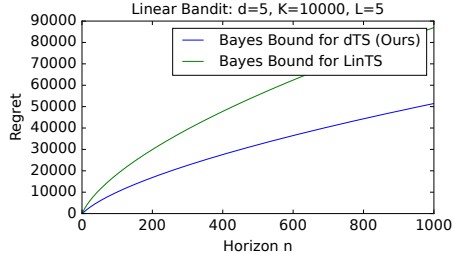

(a) Comparing our bound to the frequentist bound of `LinTS` in Abeille and Lazaric [2].

(b) Comparing our bound to the standard Bayesian bound of `LinTS`.

Figure 6: Comparing our bound to the frequentist and Bayesian bounds of `LinTS`.

## H  Extensions

**Theory beyond linear-Gaussian.** Extending our analysis to nonlinear settings is nontrivial. A promising direction is an information-theoretic analysis of Bayesian regret with structured priors [59, 52] or a PAC-Bayesian treatment similar to that used in offline contextual bandits [9]. Closing the gap to lower bounds remains open: the only known Bayesian lower bound applies to $K$-armed bandits [45], while minimax frequentist bounds scale as $\Omega(d\sqrt{n})$ [21].

$K$**-independent regret.** In the setting of Remark 2.1, where $r(x, a; \theta) = \varphi(x, a)^\top \theta$ and $\theta$ is shared across actions, our proof techniques imply $K$-independent regret once $\varphi$ is known or accurately estimated. This connects to structured large-action-space results where regret does not scale with $K$ [25, 67, 70]. Exploring further the use of diffusion models in such setting is promising.

**Robustness and misspecification.** `dTS` may face misspecification at both the prior and likelihood levels. When the diffusion prior is biased or trained on limited data, it remains empirically stable but lacks robustness guarantees. Moreover, `dTS` assumes a generalized linear reward model; deviations from this assumption leads to model misspecification.

**Beyond contextual bandits.** The posterior derivations of `dTS` extend naturally to other settings. For instance, in off-policy learning, since `dTS` defines a tractable posterior over reward parameters, it can be combined with off-policy estimators and policy improvement methods in structured offline contextual bandits environments [10].

**Online fine-tuning and offline RL.** Pre-training a diffusion model on offline data and refining it online via `dTS` amounts to diffusion fine-tuning from implicit bandit feedback. Extending this to sequential decision-making with dynamics aligns with recent diffusion-for-decision work. A concrete next step is to use `dTS` for fine-tuning pre-trained diffusion models on collected reward data.

# I   Broader impact

This work contributes to the development and analysis of practical algorithms for online learning to act under uncertainty. While our generic setting and algorithms have broad potential applications, the specific downstream social impacts are inherently dependent on the chosen application domain. Nevertheless, we acknowledge the crucial need to consider potential biases that may be present in pre-trained diffusion models, given that our method relies on them.

# J   Amount of computation required

Our experiments were conducted on internal machines with 30 CPUs and thus they required a moderate amount of computation. These experiments are also reproducible with minimal computational resources.

