# OpenReview forum: "Diffusion Models Meet Contextual Bandits"
_NeurIPS.cc/2025/Conference — NeurIPS 2025 poster_

### Official Review · Reviewer_8Fbo · 2025-06-27

**Clarity:** 2
**Significance:** 3
**Originality:** 3
**Rating:** 4
**Confidence:** 3

**Summary:**

The authors consider contextual Thompson sampling for a diffusion reward model. The reward model consists of a (diffusion) hierarchy of latent parameters, each a multivariate normal conditioned on the previous latent with mean given through a (non-linear) link function of the previous latent, followed by a head that independently maps to an action vector for each of the arms of the bandit. The reward is assumed to be a function of the alignment between a provided context vector and the unknown action vectors of each arm, following a generalized linear model.

The main contribution of this work is a tractable approximation of the posterior action vectors (and latents) in case the link function is non-linear (otherwise the diffusion model marginalizes to a Gaussian). The authors adopt the Laplace approximation for the generalized linear model and approximate the link functions as linear for the posterior update.

**Questions:**

- Concerning notation, I would suggest to only add an asterisk for the optimal action selection / policy, and a special font for random variables (or upper-case lower-case notation). Moreover, the paper would be much easier to parse if instead of defining quantities like $\mathcal L_l(\psi_l) := \mathbb P(H_t | \psi_{*,l} = \psi_l)$ the authors could just stick to writing the right-hand side expression in the formulas. Same with $Q_{t,l}$. It is barely any longer, easily fits in the space provided, and results in immediately recognizable posterior update formulas. Also, $P$ is used to denote two distinct probability distributions.

- The authors adopt a latent parameter $\psi_{1}$ through which correlation of the action vectors $\theta_{I}$ is defined. It could greatly help the reader if upon introduction of the latent parameters it was explicitly stated that the action vectors are independent conditioned on the latent.

- The histories $H_t$ are taken as a tuple with index $k \in [t-1]$ which is an unfortunate and confusing choice, because $K$ already denotes the number of arms of the contextual bandit. Instead of using $I \in [K]$ and $k \in [t-1]$, I would suggest the authors to consistently use lower and upper case, respectively, to denote matching ordinal and cardinal quantities.

- $\mathcal R^{ACT}(n)$ and $\mathcal R^{LAT}_l$ are discussed but nowhere defined in the Bayesian regret bound.

- The Hessian in Equation (3) has a typo, $g$ should have a “double-dot” to indicate the second order derivative.

- Under the linearization assumption in the theoretical analysis dTS becomes exact Thompson sampling and the prior action vectors simply become Gaussian. Doesn’t this suffice to immediately deduce regret bounds by borrowing from the extended analysis done on GP-TS? It would be great to outline the strategy for deriving the regret bounds, even outside the Appendix.

**Ethical Concerns:**

["NO or VERY MINOR ethics concerns only"]

**Final Justification:**

The authors addressed the initially unclear notation and added further experiments, in particular, an absolutely essential accuracy-compute comparison against MCMC, which demonstrates that their method is pareto-optimal. Thus I improved my clarity assessment and the final score.

**Limitations:**

yes

**Quality:**

3

**Strengths And Weaknesses:**

Strengths:
- The paper conducts both theoretical and experimental analysis for their method. The theoretical analysis is based on assuming linear link functions, where the adopted approximation becomes exact. There they inherit the cumulative regret bounds known for Thompson sampling.
- The experimental analysis is based on appropriate statistical metrics, including significance. On the Swiss-roll dataset and MovieLens, the authors demonstrate benefits of using a non-uniform prior through pre-training of the diffusion model.
- Moreover, the authors provide insights on a bias-variance trade off in the number of layers: shallow diffusion models are not expressive enough for the task at hand, whereas deep diffusion models result in excessive capacity and slow exploration.

Weaknesses:
- From the main text it is not entirely clear whether the theoretical analysis just amounts to computing the covariance matrix of the action space and plugging into Thompson sampling bounds from the standard literature, or whether a more involved analysis takes place.

- Across the paper, the adopted notation is questionable, making the paper unnecessarily difficult to read. For instance, random variables are denoted with subscript asterisks all over the place, and likelihood functions are introduced with specific notation never to be used again. See questions for suggestions on the notation.

- The experimental analysis is limited though promising. The results from Section 5.1 are entirely expected with the benchmark designed according to their method, and as far as I can tell mainly argue that in a calibrated setting adopting dTS is better than assuming independent arms. Also, the paper may benefit significantly from additional experiments in the style of MovieLens, mainly to convince practitioners that pre-training Diffusion models prior to playing bandits can give significantly better cumulative regret in the online setting.

- Since the paper introduces an efficient albeit biased approximation to posteriors in diffusion models, a significant missing piece in the experimental analysis is comparison against unbiased Monte Carlo integration. In particular, a plot with compute resources on the x-axis and regret on the y-axis for different integration budgets of Monte Carlo as well as dTS is absolutely vital for selling the method, assuming dTS is indeed pareto-optimal. This is a crucial missing point, vitally affecting the usefulness of the paper.

---

> ### Author Rebuttal · Authors · 2025-07-31
>
> We thank Reviewer **8Fbo** for their detailed and constructive feedback. We appreciate the recognition of our theoretical and empirical contributions and your suggestions for improving clarity and notation. Below, we summarize our main updates in **Version 2** (updated PDFs or external links are not supported in NeurIPS rebuttals) and address each comment in turn.
>
> ---
>
> ## Main Revisions in Version 2
>
> 1. **Expanded Experiments.**
>
>    * **MCMC vs. dTS:** Added direct comparisons showing that dTS achieves substantially lower regret and faster convergence than both “fast” (low-compute) and “slow” (high-compute) MCMC baselines.
>    * **Additional Datasets:** Beyond MovieLens and Swiss-roll, we now include MNIST and CIFAR. With only 5 % offline data, dTS yields up to 92.18 % regret improvement over LinTS and 79.56 % over HierTS on CIFAR, with similar gains on MNIST.
>
> 2. **Clarified Theoretical Analysis (Sec 3).**
>    We now explicitly outline our regret-bound derivation strategy, clarifying that it is **not** a simple covariance plug-in. Instead, we exploit the diffusion model’s **hierarchical structure** and track **prior-influence decay** via the diffusion ODE to decompose regret into **L+1 interpretable terms**, each corresponding to a latent layer of the diffusion model.
>
> 3. **Notation & Typos Fixed.**
>    We revised our notation. Thank you for these suggestions.
>
> ---
>
> ## Point-by-Point Responses
>
> **1. Missing MCMC comparison.**
> We agree that an MCMC baseline is essential. We compared dTS to two MCMC variants:
>
> | Baseline      | Regret Improvement (%) | Time Speed-up (%) |
> | ------------- | ---------------------: | ----------------: |
> | dTS-MCMC-Fast |                 50.6 % |            47.6 % |
> | dTS-MCMC-Slow |                 12.7 % |            80.5 % |
> where:
> * **Regret Improvement** = 100 × (Regret\_baseline – Regret\_dTS) / Regret\_baseline
> * **Time Speed-up** = 100 × (Time\_baseline – Time\_dTS) / Time\_baseline
>
> Even the high-compute “Slow” MCMC variant has higher regret than dTS. Applying MCMC with diffusion priors in an **online bandit** setting proved highly challenging, this was actually our first attempt and motivated our efficient approximation, which we believe is more practical.
>
> ---
>
> **2. Additional real-world datasets**
>
> > *“Beneficial to include more experiments like MovieLens to convince practitioners.”*
>
> * **Response:** We add MNIST and CIFAR benchmarks (very common benchmarks in practice). With priors trained on just **5 %** of offline data, dTS outperforms both HierTS and LinTS:
>
> | Dataset | vs. HierTS | vs. LinTS |
> | :-----: | ---------: | --------: |
> |  MNIST  |    42.17 % |   82.30 % |
> |  CIFAR  |    79.56 % |   92.18 % |
>
> We also include **cold-start ablations** where we vary the amount of offline data available to pre-train the diffusion model prior:
>
> | Offline Data (%) | vs. HierTS | vs. LinTS |
> | :--------------: | ---------: | --------: |
> |        1 %       |    69.11 % |   87.74 % |
> |        5 %       |    79.56 % |   92.18 % |
> |       25 %       |    80.65 % |   92.48 % |
> |       50 %       |    81.67 % |   92.88 % |
>
> These results demonstrate the **practical value** of pretraining diffusion priors with minimal data.
>
> ---
>
> **3. Depth of theoretical analysis**
>
> > *“Unclear if regret bound is mere covariance plug-in or more involved.”*
>
> * **Response:** * **Response:** We clarified in the revised version that while the diffusion model collapses to a Gaussian when everything is linear-Gaussian, our analysis does **not** simply compute a prior covariance and plug it into standard TS bounds. Instead, we:
> * Derive bounds from scratch,
> * Decompose the regret into **L+1 interpretable terms**,
> * Track **prior influence decay** across layers, exploiting the **ODE structure**.
>
> This yields bounds that capture **how diffusion model depth and structure** affect regret, beyond what a covariance plug-in would reveal.
>
> ---
>
> **4. Notation concerns**
>
> > *“Notation is confusing: random-variable fonts, duplicate symbols, tuple indices, undefined terms.”*
>
> * **Response:** We improved notation per your suggestions.
>
> ---
>
> Thank you again for your insightful comments. We believe that we fully addresses your concerns on empirical validation, theoretical clarity, and notation.
>
> If you feel your main concerns have been resolved, we would greatly appreciate your renewed support. Please let us know if anything remains unclear.

---

> > ### Comment · Reviewer_8Fbo · 2025-08-01
> >
> > Thank you for your very extensive additional empirical results, in particular the comparison against MCMC that demonstrates Pareto-optimality. Also, thank you for clearing up my questions and improving the notation in the paper. In light of the additional data and notational adjustments made, I will raise my score.

---

> > > ### Author Response · Authors · 2025-08-01
> > > **Sincere thanks**
> > >
> > > Thank you for your feedback and openness to updating the score. Your review has meaningfully improved our paper. We appreciate your time and engagement.

---

### Official Review · Reviewer_uoU2 · 2025-06-30

**Clarity:** 3
**Significance:** 3
**Originality:** 3
**Rating:** 4
**Confidence:** 3

**Summary:**

This paper introduces an approach to contextual bandits by leveraging pre-trained diffusion models as priors to capture complex correlations between actions, which is particularly useful when dealing with large action spaces. The authors develop diffusion Thompson sampling (dTS), a computationally efficient algorithm that approximates posterior distributions under diffusion priors through a hierarchical sampling scheme, enabling better exploration by sharing information across related actions. They provide theoretical analysis showing improved regret bounds in the linear case and demonstrate through experiments that dTS significantly outperforms standard baselines like LinUCB and LinTS, especially as the number of actions increases, while maintaining computational efficiency even with model misspecification.

**Questions:**

- Could you provide either theoretical analysis or of your two-layer approximation scheme? Specifically, I'd like to understand: (a) under what conditions the Gaussian approximation of the GLM likelihood is accurate, (b) how errors propagate through the hierarchical structure when using nonlinear link functions?

- The paper claims computational advantages, but doesn't provide wall-clock time comparisons. Could you benchmark the actual runtime of dTS versus LinTS on problems with varying K and L?

- The algorithm seems to require good offline parameter estimates to pre-train the diffusion model. How does dTS perform in a "cold start" scenario with minimal or no offline data?

**Ethical Concerns:**

["NO or VERY MINOR ethics concerns only"]

**Final Justification:**

The author has addressed my questions; therefore, I increase my score accordingly.

**Limitations:**

- The significant gap between theoretical guarantees (linear case only) and practical algorithm (nonlinear approximations) should be prominently discussed.

- The computational overhead for real-time decision-making systems needs acknowledgment.

- The lack of formal guarantees on approximation quality and failure modes.

**Paper Formatting Concerns:**

NA.

**Quality:**

3

**Strengths And Weaknesses:**

Strengths:

- The authors develop a two-stage approximation that first handles the nonlinear reward distribution by approximating the likelihood with a Gaussian centered at the maximum likelihood estimate, then addresses nonlinear link functions by borrowing the covariance update structure from the exact linear case - this design ensures the approximate posteriors reduce to exact ones in linear settings while remaining computationally tractable for general nonlinear diffusion models.
- The algorithm exploits the Markov structure of the diffusion model to avoid maintaining joint distributions over all action parameters, instead decomposing the problem into separate posterior updates for each latent layer and action - this reduces the computational burden from cubic in the total number of action parameters to linear in the number of actions plus layers.

Weaknesses:

- The theoretical analysis only covers linear diffusion models with linear-Gaussian rewards, yet the main algorithmic contribution involves two layers of approximations for nonlinear cases.

- The approach relies on pre-training diffusion models on accurate offline estimates of action parameters, but obtaining such estimates for thousands of actions is precisely the challenging problem that online learning aims to solve - the paper doesn't address how to handle settings where offline data is biased, sparse, or completely unavailable for certain actions.

- The experimental section shows performance degrades significantly when the true generative process doesn't match the diffusion prior, yet there's no theoretical characterization of robustness - the approximation scheme could amplify model misspecification errors through the hierarchical structure, but the paper doesn't analyze worst-case performance or provide guidance on detecting when the diffusion prior is misleading the algorithm.

---

> ### Author Rebuttal · Authors · 2025-07-31
>
> We thank Reviewer **uoU2** for the careful review and for emphasizing our work’s strengths in novelty, clarity, theoretical insights, and empirical performance. Since NeurIPS does not support updated PDFs or external links, all changes are detailed directly below.
>
> ---
>
> ## Summary of Key Revisions
>
> 1. **Wall-Clock Runtime Benchmarks.**
>    We added end-to-end timing comparisons demonstrating that dTS scales linearly in both K (number of actions) and L (diffusion depth), yielding orders-of-magnitude speedups over joint-parameter LinTS.
>
> 2. **Cold-Start Robustness Experiments.**
>    We introduced new ablations on CIFAR (and other benchmarks), varying the fraction of offline data used to pre-train the diffusion prior from 1 % to 50 %, and show that even with just **1 %** pre-training samples, dTS consistently outperforms LinTS and HierTS.
>
> 3. **Clarified Theory–Practice Connection.**
>    We expanded our discussion of how dTS’s posterior approximation is directly inspired by the fully linear case, and we show strong empirical evidence supporting this approximation across diverse settings.
>
> ---
>
> ## Detailed Responses**
>
> ### 1. Theoretical Analysis of the Two-Layer Approximation
>
> > **“Could you provide theoretical analysis of the Gaussian approximation and error propagation?”**
>
> Formal proofs for nonlinear link functions exceed this paper’s scope. Instead, we support our approximation with:
>
> * **Empirical Posterior Validation.** In Appendix F, we compare dTS samples to the exact posterior in a tractable linear-Gaussian case. Results show that dTS's approximation closely matches the exact posterior. This is the only case where we have access to an exact posterior to which we can compare our approximation. To futher support our approximation, we compared to two MCMC variants in the non-linear MovieLens experiment:
>
>   |          Baseline | Regret Improvement (%) | Time Speed-Up (%) |
>   | ----------------: | ---------------------: | ----------------: |
>   | dTS vs. MCMC-Fast |                 50.6 % |            47.6 % |
>   | dTS vs. MCMC-Slow |                 12.7 % |            80.5 % |
>
>   Even the high-compute MCMC variant yields higher regret than dTS, motivating our efficient approximation for online bandits.
>
> * **Strong Empirical Performance.** We added MNIST and CIFAR benchmarks (common practice datasets). With just **5 %** offline data, dTS outperforms HierTS and LinTS:
>
>   | Dataset | vs. HierTS | vs. LinTS |
>   | ------: | ---------: | --------: |
>   |   MNIST |     42.2 % |    82.3 % |
>   |   CIFAR |     79.6 % |    92.2 % |
>
> * **Design-Principle Alignment.** Our approximation reduces to the exact posterior when the diffusion model is linear and the reward model is Gaussian. As more data accrue, the prior’s influence naturally diminishes; this behavior matches the exact posterior distribution behavior, and hence we believe it is a strong point of our approximation.
>
> ### 2. Role of Linear-Case Guarantees
>
> > **“How do linear-case guarantees inform the nonlinear algorithm?”**
>
> Our linear analysis inspired our approximation for two reasons:
>
> 1. **Exact-Case Posterior Approximation.** dTS matches the linear-case posterior when models are linear. Thus, the linear-case posterior fully inspired our posterior approximation in the non-linear case.
> 2. **Regret-Scaling Alignment.** Once the prior captures the true distribution, dTS’s regret scaling with L matches the linear-case scaling predicted by our theory (Figure 4 (b)).
>
> ### 3. Wall-Clock Runtime Comparisons
>
> > **“Please benchmark actual runtimes of dTS vs. LinTS.”**
>
> We measured average runtimes over 100 GPU runs:
>
> * **Varying K (L=2, d=5):**
>
>   |        Method |   K=200 |     400 |      800 |      1600 |
>   | ------------: | ------: | ------: | -------: | --------: |
>   |       **dTS** |  1.95 s |  3.78 s |   7.39 s |   14.82 s |
>   | LinTS (joint) | 17.13 s | 98.88 s | 611.61 s | 3834.61 s |
>
> * **Varying L (K=200, d=5):**
>
>   |        Method |     L=4 |       8 |      16 |      32 |
>   | ------------: | ------: | ------: | ------: | ------: |
>   |       **dTS** |  3.73 s |  3.78 s |  3.92 s |  4.05 s |
>   | LinTS (joint) | 95.05 s | 95.73 s | 96.08 s | 95.84 s |
>
> These results confirm dTS’s **linear** scaling versus LinTS (joint) **cubic** cost, making dTS more practical at large scales.
>
> ### 4. Cold-Start Performance with Minimal Offline Data
>
> > **“How does dTS perform with sparse or no offline data?”**
>
> In Sec 5.4 and Fig 4(a), we show that with only **10 pre-training samples**, dTS outperforms LinTS on the Swiss-roll benchmark. Our additional CIFAR ablations yield:
>
> | Offline Data (%) | vs. HierTS | vs. LinTS |
> | ---------------: | ---------: | --------: |
> |              1 % |     69.1 % |    87.7 % |
> |              5 % |     79.6 % |    92.2 % |
> |             25 % |     80.7 % |    92.5 % |
> |             50 % |     81.7 % |    92.9 % |
>
> These results confirm that dTS **effectively leverages limited prior data** to capture action correlations and achieve substantial regret reductions.
>
> ---
>
> We believe these revisions significantly strengthen our theoretical grounding, empirical validation, and practical utility. If these changes address your concerns, we kindly request your renewed support for our submission.

---

> ### Author Response · Authors · 2025-08-01
> **Brief clarification on metrics and compute**
>
> We wanted to add this brief comment to further clarify our rebuttal.
>
> * **Metrics.** All percentages denote improvements defined as:
>
>   * **Regret improvement:** 100 x (Regret\_baseline − Regret\_dTS) / Regret\_baseline.
>   * **Spped-up improvement:** 100 x (Time\_baseline – Time\_dTS) / Time\_baseline.
>
> * **Wall-Clock. Runtime** Our paper claims that dTS is significantly faster than the LinTS variant that models action parameters jointly. This is why, in the requested wall-clock experiments, we compared dTS to LinTS (joint) to validate our claim. Also, runtime of dTS increases slightly as $L$ grows; this increase is small in practice (code included in the submission) and only noticeable at larger $L$.

---

> > ### Author Response · Authors · 2025-08-05
> > **We Are Happy to Engage in a Discussion**
> >
> > Dear Reviewer uoU2,
> >
> > Thank you again for your constructive feedback. We are happy to engage in further discussion and, to facilitate this, would like to briefly summarize the key points from our rebuttal:
> >
> > * **W1-Q1 (Non-linear theory):** As we clarified in our rebuttal, a full non-linear analysis is beyond this paper's scope (a common approach in bandits is to analyze an algorithm in the linear setting despite its general applicability). However, we provided **strong justification for our approximation:** It is exact in the linear case, matches posterior behavior in limiting data regimes, and **empirically outperforms** both MCMC-based dTS and standard bandit baselines.
> >
> > * **Q2 (Computational Efficiency):** In our rebuttal, we added wall-clock time experiments demonstrating the favorable scaling of our method with respect to actions ($K$) and features ($L$).
> >
> > * **W2-Q3 (Cold-Start / Limited Data):** Our rebuttal included new ablation studies for extreme cold-start scenarios. These results complement our existing Fig. 4a and show that dTS remains robust even when the initial diffusion model is poorly trained.
> >
> > * **W3 (Model Misspecification):** As we highlighted from our experiments (Fig. 4a, 3c), performance degrades gracefully and not significantly. Even with a poorly specified model, dTS **still outperforms strong baselines** like LinTS.
> >
> > We hope this summary is helpful. If any specific concern remains that prevents you from a more positive assessment, we would greatly appreciate knowing what it is. We are ready to provide any further clarification.

---

### Official Review · Reviewer_oesN · 2025-07-02

**Clarity:** 4
**Significance:** 3
**Originality:** 3
**Rating:** 5
**Confidence:** 3

**Summary:**

* This paper proposes a new contextual bandit algorithm that extends the literature on hierarchical Bayesian bandits to a more complex structure where the prior distribution is represented as a pre-trained diffusion model.

* This paper considers the contextual bandit setting with prior represented as a diffusion model, and adopts Bayes regret to measure the performance. Then, it proposes a novel algorithm - Diffusion contextual Thompson sampling (DTS).
    * The first key novelty is deriving an approximation for the posterior update. This approximation has good properties: (1) they match the prior with no data, and the prior's effect diminishes as data accumulates. (2) The approximate posterior is also a diffusion model with updated means and covariances.
    * The second key contribution is to theoretically analyze the Bayes regret bound in the simplified setting where the known diffusion model's ODE is time-varying linear–Gaussian. The challenge of analysis the recursive update in the diffusion model. The analysis demonstrates the computational and statistical benefits of the proposed method, in contrast to LinTS [54]. The key insight is that the proposed method leverages the correlation between actions through the pre-trained diffusion model, while LinTS [54] has to learn each action independently.

* Empirical analysis contains two experiments:
  * When the prior matches the generative process, the paper compares the proposed method against LinUCB [1], LinTS [3], and HierTS [31] for the linear-reward setting, and against UCB-GLM [44] and GLM-TS [16] for the nonlinear-reward setting. The result shows that the proposed method achieves lower regret than the baselines. It demonstrates the power of a well-specified prior.
  * When the prior does not match the generative process, such as Swiss-roll data and MovieLens data, the proposed method also outperformed LinTS.

**Questions:**

* In Sec.4.1, the paper compares the proposed method vs LinTS [3]. I am not very familiar with [3], but it seems that the paper [3] (https://arxiv.org/pdf/1209.3352) 's Sec.2.1 assumes that there is a common reward parameter µ, which seems different from this paper's setting where each action `i' has its own parameter `θ_{*,i}`. I am curious whether you think it would still be appropriate to compare the proposed method with [3]?
* Another question I have is that Fig. 2's nonlinear diffusion linear reward case, why not also benchmark the HierTS [31] method?
* Out of curiosity, this paper focuses on correlations between actions. What about also exploiting correlations between contexts? E.g., similar users would have similar preferences about movies. Would the proposed method also extend to handle that?

**Ethical Concerns:**

["NO or VERY MINOR ethics concerns only"]

**Final Justification:**

This paper is well motivated, with theoretical and promising empirical results. I think it extends contextual bandit to be able to bake richer information into the action prior. The rebuttal clarified my questions. I would keep my score.

**Limitations:**

Yes

**Quality:**

3

**Strengths And Weaknesses:**

* This paper solves a well-motivated problem and is well-written.
* The algorithm is novel and sound. Both theoretical and empirical result supports the method.
* One concern is that it seems that the result section focuses on comparing the proposed method vs LinTS (a baseline that does not exploit the correlation between action features). This is very effective in showing that the proposed method can exploit the action correlation, and it is actually effective and useful. However, it would be even stronger to also demonstrate that the proposed method is better than other hierarchical Bayesian bandit methods, to demonstrate that representing the prior as a diffusion model is a useful, not an overkill, in real-world scenarios. In this way, the two key takeaway messages are proved: (1) exploiting action correlation is useful (already somewhat demonstrated in prior literature), (2) representing prior distribution as a diffusion model is useful (very novel).
    * To show this, I think there are a couple of ways. For example, for the real-world dataset - movielens, it would be helpful to benchmark the HierTS [31] method, to see how much modeling the prior as diffusion is beneficial in the real world. Alternatively, the author could take the setting of HierTS [31] and show that the diffusion-model prior allows better learning performance because the diffusion model is more expressive.

---

> ### Author Rebuttal · Authors · 2025-07-31
>
> We thank Reviewer **oesN** for the thorough review and for highlighting our paper’s clear motivation, sound algorithmic design, and strong theoretical and empirical results. Since NeurIPS disallows updated PDFs or external links, all revisions are detailed below.
>
> ---
>
> ## Summary of Key Revisions
>
> 1. **Added HierTS \[31] Baseline.**
>    We now benchmark HierTS on MovieLens and Swiss-roll. On MovieLens, HierTS incurs ≈ 45.5 % higher cumulative regret than dTS, with dTS > HierTS > LinTS in terms of performance.
>
> 2. **Expanded Nonlinear-Reward Evaluation.**
>    Fig. 2’s nonlinear-diffusion, linear-reward experiments now include HierTS, where dTS still outperforms it.
>
> 3. **Clarified LinTS \[3] Implementation.** We run LinTS per action (one Gaussian posterior per \$\theta\_{\*,i}\$), matching our setting. We cite \[3, Sec. 2.1] and per-arm TS variants. Joint-parameter variants would require modeling the full \$Kd\$-dimensional posterior, which is computationally prohibitive.
>
> 4. **Context-Correlation as Future Work.**
>    We agree that modeling correlations between contexts (e.g., similar users) is valuable. While this paper focuses on action correlations, extending diffusion priors to contexts is an exciting direction for future research.
>
> ---
>
> ## Point-by-Point Responses
>
> **1. Comparison to other hierarchical Bayesian bandits (HierTS \[31])**
> Reviewer: *How does dTS compare to HierTS in MovieLens?*
> Despite using the same offline data, HierTS’s Gaussian hyperprior yields up to **45.5 %** higher regret than dTS’s diffusion prior in MovieLens, demonstrating that diffusion models capture richer action correlations. We also conducted additional MNIST and CIFAR benchmarks. With just **5 %** offline data, dTS outperforms both HierTS and LinTS:
>
> | Dataset | vs. HierTS | vs. LinTS |
> | :-----: | ---------: | --------: |
> |  MNIST  |     42.2 % |    82.3 % |
> |  CIFAR  |     79.6 % |    92.2 % |
>
> **2. Exploiting correlations between contexts**
> Reviewer: *Can dTS handle context correlations (e.g., user similarity)?*
> We agree this is an important extension. Prior work (e.g., clustering-based Thompson sampling) has explored simple context correlations, but diffusion-based context priors remain unexplored. We plan to investigate this direction in future work.
>
> ---
>
> We believe these updates address your concerns and further demonstrate the practical advantages of diffusion-model priors. Thank you again for your valuable feedback; we welcome any additional comments or suggestions.

---

> ### Author Response · Authors · 2025-08-01
> **Brief clarification on percentage metrics**
>
> We wanted to add this brief comment to further clarify that all percentages included in the rebuttal denote improvements defined as:
>
>   * **Regret improvement:** 100 x (Regret\_baseline − Regret\_dTS) / Regret\_baseline.

---

> > ### Comment · Reviewer_oesN · 2025-08-04
> >
> > Thank you so much for the response! It clarifies my questions. I intend to keep my score.

---

### Official Review · Reviewer_taUg · 2025-07-03

**Clarity:** 3
**Significance:** 3
**Originality:** 3
**Rating:** 4
**Confidence:** 2

**Summary:**

This paper leverages diffusion models to tackle the challenge of decision-making in contextual bandits with large and complex action spaces. The authors propose a framework that uses pre-trained diffusion models as expressive priors within a Thompson Sampling (TS)-style bandit algorithm. Their proposed method enables efficient posterior approximation and sampling by combining offline-trained diffusion models with online updates during bandit online interactions. The authors develop practical implementations for posterior inference under this setup, and conduct experiments to demonstrate strong empirical performance across a range of contextual bandit scenarios. The authors provide theoretical analysis for the linear case, where the regret of the proposed method is bounded.

**Questions:**

1. How can the proposed method extend to more general nonlinear settings?
2. How can the proposed method extend to the reinforcement learning settings?
3. Your work proposes a TS-type algorithm. Do you think the idea of leveraging diffusion models to enhance contextual bandits can also apply to the UCB-type algorithmic approach?

**Ethical Concerns:**

["NO or VERY MINOR ethics concerns only"]

**Final Justification:**

The authors have addressed my concerns, and I believe this is a good work with solid contributions

**Limitations:**

Yes

**Quality:**

3

**Strengths And Weaknesses:**

Strengths:
1. This paper studies an important problem of contextual bandits. To the best of my knowledge, this is the first work to leverage diffusion models to enhance contextual bandits. The paper is well-motivated.
2. The paper is well-written and easy to follow.
2. The authors provide theoretical guarantees of the proposed method.
3. Extensive experiments are conducted to show the outperformance of the proposed methods.

Weaknesses:
1. This paper considers the linear case, which is a slightly restrictive assumption in practice, but it is reasonable in the bandit literature.
2. There are many works on diffusion models for reinforcement learning, though mostly from the empirical perspective. It would be better to discuss these works and make some comparisons. For example, see the related works in the survey "Diffusion Models for Reinforcement Learning: A Survey".
3. The authors conducted extensive simulations, but they are only on synthetic and MovieLens datasets. It would be better to conduct experiments on more datasets to show the consistent improvement of the proposed methods over existing ones.

---

> ### Author Rebuttal · Authors · 2025-07-31
>
> We thank Reviewer **taUg** for your thoughtful review and your positive remarks on our motivation, clarity, theoretical contributions, and empirical results. Below, we summarize the main updates made (in this rebuttal only, as external links are disallowed) and respond to your comments point-by-point.
>
> ---
>
> ## Summary of Key Revisions (Version 2)
>
> 1. **Expanded Related Work**
>    We added discussion contrasting our *online contextual bandit* setting with recent *diffusion-RL* works and cite the 2024 survey *Diffusion Models for Reinforcement Learning* along with key references therein.
>
> 2. **New Experiments on MNIST & CIFAR**
>    Added standard supervised-to-bandit benchmarks. With just **5 %** offline data for pretraining, dTS yields:
>
>    | Dataset | vs. HierTS | vs. LinTS |
>    | ------: | ---------: | --------: |
>    |   MNIST |    42.17 % |   82.30 % |
>    |   CIFAR |    79.56 % |   92.18 % |
>
> 3. **Cold-Start Ablations**
>    We include ablations across 1–50 % offline data. dTS consistently outperforms all baselines:
>
>    | Offline Data (%) | vs. HierTS | vs. LinTS |
>    | ---------------: | ---------: | --------: |
>    |              1 % |    69.11 % |   87.74 % |
>    |              5 % |    79.56 % |   92.18 % |
>    |             25 % |    80.65 % |   92.48 % |
>    |             50 % |    81.67 % |   92.88 % |
>
> 4. **Clarified Nonlinear Reward Support**
>    dTS supports any twice-differentiable reward function via Laplace approximation. The linearity assumption is used solely for deriving theoretical regret bounds.
>
> 5. **Limitations Moved from Appendix to Main Text**
>    The limitations section has been elevated from the appendix for improved visibility.
>
> ---
>
> ## Point-by-Point Responses
>
> ### 1. Linear-Case Theory
>
> > **“The linear assumption is slightly restrictive.”**
>
> Our theoretical analysis is limited to the linear setting (common in bandit theory for tractability), but the algorithm itself supports non-linear rewards and non-linear diffusion models via our approximation. Our empirical results on high-dimensional tasks (e.g., CIFAR) confirm strong performance.
>
> ---
>
> ### 2. Diffusion-RL Discussion
>
> > **“There are many works on diffusion models for RL.”**
>
> We cite relevant papers including the 2024 survey. Our setting is distinct in that we focus on *exploration in online contextual bandit* via posterior sampling, not offline policy optimization.
>
> ---
>
> ### 3. Dataset Diversity
>
> > **“It would be better to conduct experiments on more datasets.”**
>
> We agree and added experiments on **MNIST** and **CIFAR**, which are common benchmarks in contextual bandits. These show large and consistent improvements over LinTS and HierTS, even with minimal offline data. See results in the Summary of Key Revisions section above.
>
> ---
>
> ### 4. Q1 – Extension to Nonlinear Settings
>
> > **“How can the method extend to more general non-linear settings?”**
>
> This is already supported: our approximation handles any non-linear twice-differentiable reward (e.g., logistic) and any non-linear diffusion model. We clarified this and expanded examples in App C.
>
> ---
>
> ## Extensions
>
> __RL Extension.__
>
> We agree this is a promising direction. A natural extension is to use dTS for **model-based RL**, where policy or transition dynamics parameters are drawn from a diffusion prior and updated online via posterior sampling. We look forward to exploring this idea further.
>
> __UCB-Type Extension.__
>
> In Appendix J, we added a sketch for **BayesUCB-style variant**, where the selected action maximizes the posterior mean plus a bonus term based on posterior uncertainty (computed using the posterior covariance). We believe this could yield promising performance and plan to explore this direction in future work.
>
> ---
>
> We hope these additions, more datasets, deeper context, and clarification of practical extensions, address your concerns and strengthen your confidence in our submission. We would greatly appreciate your consideration for raising your support of our paper.

---

> > ### Author Response · Authors · 2025-08-01
> > **Brief clarification on percentage metrics**
> >
> > We wanted to add this brief comment to further clarify that all percentage metrics denote regret improvements defined as:
> >
> >   * **Regret improvement:** 100 x (Regret\_baseline − Regret\_dTS) / Regret\_baseline.

---

> ### Comment · Reviewer_taUg · 2025-08-05
>
> Thanks for the response, I will keep my score.

---

### Note · Authors · 2025-08-12

We thank the Area Chair and reviewers for their time and constructive feedback. Below is a concise, factual summary of our discussions.

**Contribution claim.** We propose **diffusion Thompson sampling (dTS)**, which uses a pre-trained diffusion prior to better model the distribution of action parameters and share information across arms. Our main algorithmic innovation is a **computationally efficient posterior approximation** that is fast to update and sample from online. This approximation is principled:

* It becomes **exact** in the linear–Gaussian case, for which we provide a **Bayes-regret analysis**.
* It preserves key asymptotics of the exact posterior: it matches the prior with no data, and the prior’s influence diminishes as data accumulate.

**Evidence in the paper.**

* **Theory (linear–Gaussian case):** We provide a Bayes-regret bound highlighting improved scaling with the number of actions $K$.
* **Computational efficiency:** dTS is more scalable than methods maintaining a joint posterior, with time complexity $O((K{+}L)d^3)$ versus $O(K^3 d^3)$.
* **Empirical validation:** On synthetic, Swiss-roll, and MovieLens, **dTS consistently outperforms HierTS and LinTS**; the gap vs. LinTS widens as $K$ increases; performance degrades gracefully under prior misspecification. A cold-start experiment on Swiss-roll shows dTS outperforming LinTS with as few as 10 pre-training samples. In our reported experiments, **dTS > HierTS > LinTS**.

**Supporting evidence in the rebuttal.**

* **New datasets (MNIST and CIFAR):** With 5% pre-training, dTS shows substantial regret reductions on MNIST and CIFAR (consistent with dTS > HierTS > LinTS).
* **Extreme cold-start ablations:** On CIFAR, dTS remains effective with as little as 1% pre-training data.
* **MCMC & runtime benchmarks:** dTS attains lower regret than a high-compute MCMC baseline while being faster in practice, yielding a better regret–compute trade-off.
* **Wall-clock confirmation:** Confirms the time complexity predicted by our analysis.
* **Clarifications:** BayesUCB-style variant sketched; positioning relative to diffusion-RL clarified; limitations moved to the main text.

In summary, thanks to the reviewers’ feedback, the rebuttal provides clarifications and confirmatory results that reinforce the paper’s original claims and evidence, including additional datasets, cold-start sensitivity, and wall-clock behavior aligned with the stated complexity. We appreciate the committee’s consideration.

---

### Decision · Program_Chairs · 2025-09-17

**Decision:**

Accept (poster)

**Comment:**

The paper proposes a new contextual bandit algorithm that extends the literature on hierarchical Bayesian bandits to a more complex setting, where the prior distribution is represented by a pre-trained diffusion model. The authors analyze the Bayesian regret when the reward functions are linear and the prior is Gaussian. Numerical experiments are presented to illustrate the efficiency of the proposed method.

All reviewers acknowledged the novelty of the approach and the promising experimental results. However, several shortcomings were noted that should be addressed in a revised version of the paper:

•	The theoretical guarantees are derived only for the linear-Gaussian case, which essentially means that the prior is Gaussian and that the diffusion model is not actually necessary. The authors should comment on this limitation and/or explain how the analysis might be extended.

•	The use of a pre-trained diffusion model can be problematic if offline data is limited. While the authors addressed this concern in the rebuttal, the discussion should be further developed and explicitly included in the paper.

•	The experiments should be expanded and completed as suggested by Reviewer 8Fbo.

Overall, the paper introduces a novel approach to contextual linear bandits. Unfortunately, the theoretical results only support the Gaussian prior setting, essentially reducing the role of the diffusion model. Therefore, to strengthen the contribution, the paper should include extended numerical experiments that both motivate the use of diffusion priors and demonstrate their advantages in practice.